# Context-dependent signaling of coincident auditory and visual events in primary visual cortex

Thomas Deneux, Evan R Harrell, Alexandre Kempf, Sebastian Ceballo, Anton Filipchuk, Brice Bathellier*

Department for Integrative and Computational Neuroscience (ICN), Paris-Saclay Institute of Neuroscience (NeuroPSI), UMR9197 CNRS, University Paris Sud, Gif-sur-Yvette, France

**Abstract** Detecting rapid, coincident changes across sensory modalities is essential for recognition of sudden threats or events. Using two-photon calcium imaging in identified cell types in awake, head-fixed mice, we show that, among the basic features of a sound envelope, loud sound onsets are a dominant feature coded by the auditory cortex neurons projecting to primary visual cortex (V1). In V1, a small number of layer 1 interneurons gates this cross-modal information flow in a context-dependent manner. In dark conditions, auditory cortex inputs lead to suppression of the V1 population. However, when sound input coincides with a visual stimulus, visual responses are boosted in V1, most strongly after loud sound onsets. Thus, a dynamic, asymmetric circuit connecting AC and V1 contributes to the encoding of visual events that are coincident with sounds.
DOI: https://doi.org/10.7554/eLife.44006.001

*For correspondence:
brice.bathellier@cnrs.fr

Competing interests: The authors declare that no competing interests exist.

## Introduction

Numerous multisensory illusions (*Bonath et al., 2007*; *Maeda et al., 2004*; *Sekuler et al., 1997*; *Shams et al., 2000*) show that audition and vision have strong perceptual bonds. For example, in the double flash illusion (*Shams et al., 2000*) a brief sequence of two sounds played during a single visual flash leads to the perception of two flashes. While the mechanisms of cross-modal perceptual interactions remain unclear, the anatomy of the mammalian brain shows multiple sites of auditory-visual convergence. One of the best studied integrative regions is the superior colliculus, where visual and auditory cues are combined in the networks computing gaze direction (*Stein and Stanford, 2008*; *Witten and Knudsen, 2005*). In cortex, associative areas such as those in the parietal cortex (*Raposo et al., 2014*; *Song et al., 2017*) are not the only cortical sites demonstrating auditory-visual convergence: increasing evidence shows that functional auditory-visual interactions exist already in primary sensory cortex (*Ibrahim et al., 2016*; *Iurilli et al., 2012*; *Kayser et al., 2010*; *Kayser et al., 2008*; *Meijer et al., 2017*). Moreover, axonal tracing studies indicate that numerous direct connections exist between primary auditory and visual cortex (*Bizley et al., 2007*; *Driver and Noesselt, 2008*; *Falchier et al., 2002*; *Ghazanfar and Schroeder, 2006*; *Innocenti et al., 1988*; *Leinweber et al., 2017*; *Zingg et al., 2014*), suggesting that auditory-visual cross-talk is present before the associative stage. The existence of these direct connections in mice has initiated some experimental inquiry into their potential role. Recent studies indicate that auditory to visual connections are much stronger than their reciprocals, and that they provide inputs to visual cortex that can modulate visually driven activity (*Ibrahim et al., 2016*; *Iurilli et al., 2012*). Nevertheless, the computational role of primary auditory-visual connections remains unclear. First and foremost, information is lacking about the auditory features channeled from auditory to visual cortex and how they are integrated into the visual processing stream.

Mouse auditory cortex encodes a wide variety of acoustic features (*Mizrahi et al., 2014*). Spectral content (*Issa et al., 2014*; *Kanold et al., 2014*) and temporal features such as modulations of frequency (*Trujillo et al., 2011*) or intensity (*Gao and Wehr, 2015*) are the most well-studied, and intensity variations occurring at sound onsets and offsets (*Deneux et al., 2016b*; *Scholl et al., 2010*) are particularly salient auditory features. The question of how sound frequency information would map onto visual cortex is a difficult one, because of the lack of perceptual and ethological data on the particular frequency cues that could potentially be associated with particular visual stimuli in mice. In contrast, temporal coincidence is known to be used for perceptually assigning auditory and visual stimuli to the same object and is implicated in the double flash (*Slutsky and Recanzone, 2001*) and ventriloquist illusions (*Bonath et al., 2007*; *Recanzone, 2009*). Detection of temporal coincidence involves determining when sounds begin and end, and therefore might implicate neurons that encode particular intensity envelope features such as onsets and offsets. Also, covariations of the size of a visual input and sound intensity envelope are important for binding looming and receding auditory-visual stimuli (*Tyll et al., 2013*). This suggests that there could be preferential cross-talk between some intensity envelope features and visual information. We have recently demonstrated that envelope features such as onsets, offsets and sustained temporal dynamics are encoded in separate cells with further selectivity for different sound amplitudes. Some neurons respond only to high amplitude, 'loud' sound onsets, whereas others only respond to low amplitude, 'quiet' onsets, and neurons responding to offsets and sustained phases are also tuned to precise intensity ranges (*Deneux et al., 2016b*). Some neurons also encode combinations of these features. Such a coding scheme possibly allows constructing finer sound categories or interpretations. For example, mechanical shocks produce sounds that rise abruptly in intensity and tend to activate loud onset neurons, while more continuous events produce sounds that progressively ramp up in intensity and first activate quiet onset neurons. This complexity opens the possibility that some envelope features are more relevant to complement visual information related to external events. However, the precise intensity envelope features that characterize sound onsets and offsets which are transmitted through the direct connection between AC and V1 and how they impact visual processing remain unknown.

To address this question, we used two-photon calcium imaging and intersectional genetics to identify some of the intensity envelope features encoded by AC neurons that project to V1, and showed that V1-projecting neurons are predominantly tuned to loud onsets while other tested envelope features are less prominently represented, at least when comparing with supragranular cortical layers that lack V1-projecting cells. We also show that this cross-modal information impacts V1 in a context-dependent manner, with a net inhibitory effect in darkness and a net positive effect in the light, probably mediated by a subpopulation of L1 inhibitory neurons. Furthermore, we show that this mechanism also allows for a boosting of visual responses that occurs together with auditory events, particularly when those include loud sound onsets, increasing the saliency of visual events that co-occur with abrupt sounds.

## Results

### Abrupt sound onsets are a dominant feature encoded by V1-projecting neurons

To evaluate if particular on- and offset features are preferentially channeled from AC to V1, we compared the feature encoding in the overall population of AC neurons and in AC neurons projecting to the primary visual cortex. To identify these V1-projecting cells, we injected a canine adenovirus (CAV) expressing Cre, which is retrogradely transported through axons, into V1. In the same mice, we injected into AC an adeno-associated virus (AAV) expressing GCAMP6s (*Chen et al., 2013b*) in a Cre-dependent manner (*Figure 1A*). As a result, GCAMP6s expression was obtained exclusively in AC neurons that project to V1. Consistent with previous reports, we observed that these neurons were located predominantly in layer 5 (*Figure 1B*) (*Ibrahim et al., 2016*). As a comparison, in another set of mice, we injected an AAV1-syn-GCAMP6s virus to express the calcium reporter in AC neurons, independent of cell type. However, we selected cells according to their vertical localization: upper layer 5 (370–430 µm, corresponding to the depth at which V1-projecting cells were imaged) and layer 2/3 (150–250 µm depth). Using two-photon microscopy, we imaged calcium responses at

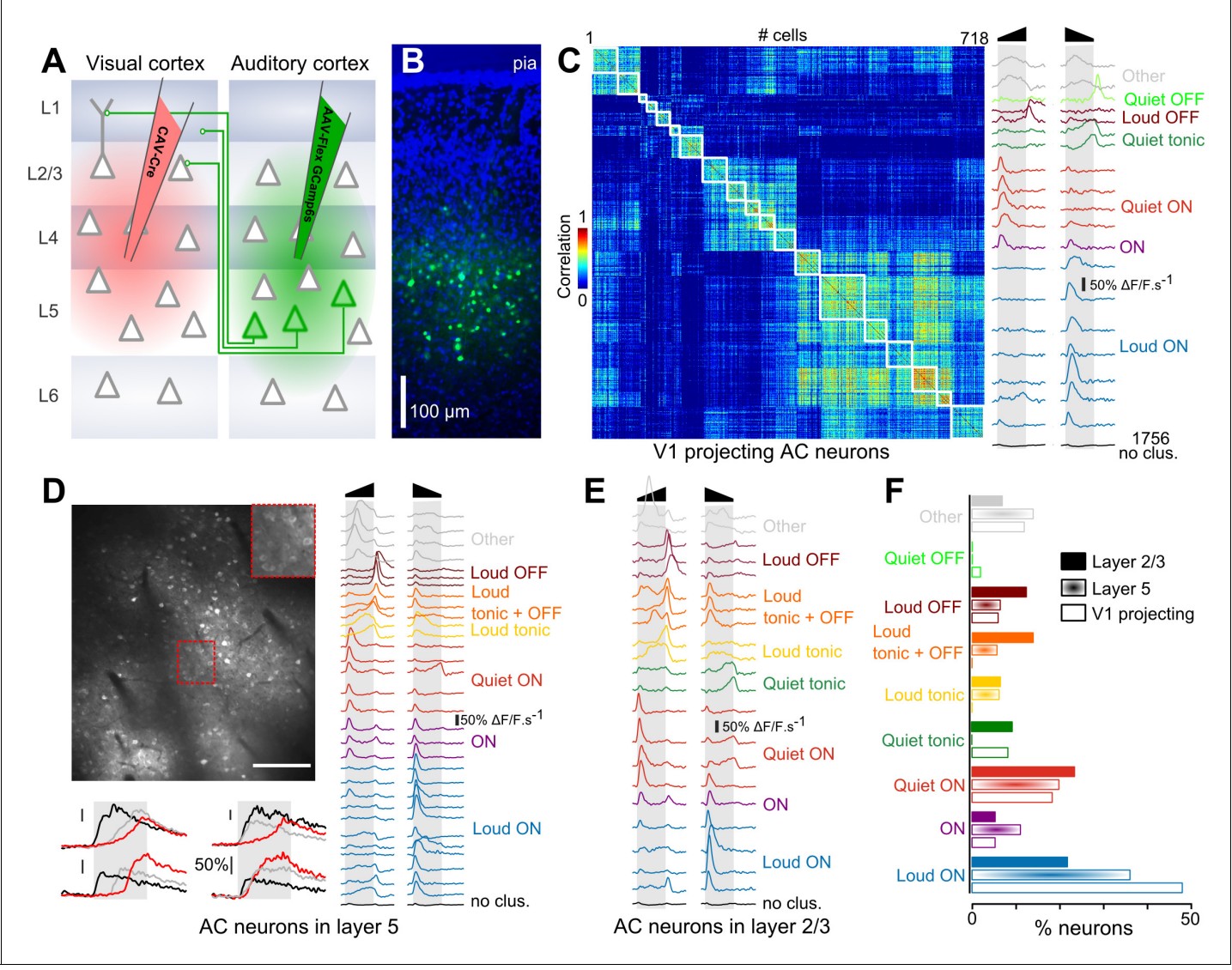

**Figure 1.** Loud onset responses dominate in V1-projecting neurons. (**A**) Viral expression strategy for GCAMP6s labeling of V1-projecting neurons in AC. (**B**) Epifluorescence image of an AC histological section (blue = DAPI, green = GCAMP6 s) showing V1-projecting neurons (cortical layers are matched to A). (**C**) Matrix of sound response correlation for all cells that could be assigned to a cluster for auditory cortex neurons that project to V1. The average response traces of each cluster (mean deconvolved calcium signal across cells) are shown on the right with a color code corresponding to the functional label given to the cluster. The bottom trace (black) shows the average response for the cells that were not assigned to a cluster (non- or weakly-responsive cells). (**D**) Top: Example of a 1 × 0.8 mm imaging field of view showing GCAMP6s-expressing neurons in upper layer 5 of auditory cortex (depth: 400 µm) with a magnification in the top-right inset. Bottom: Example raw calcium traces for four neurons during sound presentation. Red, black and gray traces are different samples for the same neuron. Scale bars 50% ΔF/F. Right: Average response traces of each cluster obtained with neurons recorded in upper layer 5 (370 to 430 µm depth). The color code corresponds to the functional label used in C. The black trace stands for non- or weakly-responsive cells. (**E**) Average response traces of each cluster obtained with neurons recorded in layer 2/3. (**F**) Fraction of neurons corresponding to each cluster shown in C-E. The functional cell class distributions are significantly different across the three anatomical cell types (L5, L2/3 or V1-projecting) as assessed together and pairwise across cell type using the $\chi^2$ test of independence (p<$10^{-64}$). The fraction of loud onset cells is significantly different ($\chi^2$ test of independence, performed pairwise across anatomical cell types p<$10^{-9}$).

DOI: https://doi.org/10.7554/eLife.44006.002

The following figure supplements are available for figure 1:

**Figure supplement 1.** Correlation matrices for all clustered neurons.

DOI: https://doi.org/10.7554/eLife.44006.003

**Figure supplement 2.** Layer 5 responses to up- and down-ramps recorded using electrophysiology.

DOI: https://doi.org/10.7554/eLife.44006.004

**Figure supplement 3.** AC neurons projecting to V1 are tuned both to intensity and frequency.

*Figure 1 continued on next page*

*Figure 1 continued*

DOI: https://doi.org/10.7554/eLife.44006.005

single cell resolution in these three sets of mice (layer 2/3: 4616 neurons, 29 sessions, 11 mice; upper layer 5: 7757 neurons, 12 sessions, four mice; V1-projecting: 2474 neurons, 15 sessions, three mice) while playing white noise sounds ramping up or down in intensity (range 50 to 85 dB SPL).

After automated ROI extraction (*Roland et al., 2017*), calcium signals were deconvolved (*Bathellier et al., 2012*; *Yaksi and Friedrich, 2006*) to obtain a temporally more accurate estimate of actual neuronal firing rates by reverse-filtering the slow decay dynamics of calcium signals and smoothed with a sliding window filter (190 ms) to reduce fast noise. The resulting averaged response profiles were then submitted to a hierarchical clustering (*Deneux et al., 2016b*) (see Materials and methods) to group together cells with similar response profiles and thereby identify different functional response types. In order to better appreciate the variety of responses, we over-clustered the data, based on the same homogeneity threshold for all datasets, so that most response types are represented by multiple clusters (*Figure 1C–E*). The clustering also identified a number of weakly or non-responsive cells that were not included in the analysis. We obtained 19, 31 and 21 functional clusters explaining 52.3 ± 7.2%, 51.9 ± 6.4% and 51.3 ± 6.5% of the total variance (mean ±STD) for V1-projecting, layer 5 and layer 2/3 cells, respectively. These high fractions of explained variance and the strong homogeneity of the clusters as seen in response correlation matrices (*Figure 1C* and *Figure 1—figure supplement 1*) indicate that the clustering provides an accurate dimensionality reduction of the dataset. Within these clusters, we identified a maximum of 9 response types (*Figure 1C–E*), including onset (ON), offset (OFF) and sustained (Tonic) responses with loud or quiet intensity tuning, and also clusters combining loud OFF and sustained responses (labeled as 'loud OFF +tonic') or quiet and loud ON responses (labeled as 'ON'), as described previously (*Deneux et al., 2016b*). Note that the terms 'loud' and 'quiet' are used here in their relative sense and correspond to high (~85 dB) and lower (~50 dB) sound pressure levels respectively, within the range used in this study. As a validation of the approach, many of these response types could also be observed in layer 5 targeted extracellular recordings in roughly the same proportion as in two-photon calcium imaging (*Figure 1—figure supplement 2*). Interestingly, the distribution of these response types was different across layer 2/3, layer 5 and V1-projecting populations (*Figure 1F*). While some slight differences could be seen for almost all response types, the largest discrepancies were observed for neurons signaling loud onsets which represented 48% of the clustered neurons in V1-projecting cells, and only 36% and 21% in layer 5 and layer 2/3, respectively (*Figure 1F*). Thus, loud onset responses were enriched in layer 5 which contains most of the cells with direct projections to V1 and predominant in cells retrogradely labeled as V1-projecting neurons, indicating that AC is organized to favor the transfer of this particular sound envelope information even if other envelope features are also conveyed by V1-projecting cells (*Figure 1C*), as well as some sound frequency information (*Figure 1—figure supplement 3*).

## The sign of auditory responses in V1 depends on the illumination context

How do these structured AC inputs impact V1? One study in which mice received no visual input (dark environment) suggested an inhibitory effect (*Iurilli et al., 2012*) while another study in which mice received visual stimuli described excitatory effects (*Ibrahim et al., 2016*). Thus, the sign of AC inputs to V1 pyramidal cells could be context-dependent. To address this question, we performed two-photon calcium imaging in head-fixed awake mice (*Figure 2A*) using the calcium sensor GCAMP6s (*Figure 2B*) expressed in V1 through stereotactic injection of an AAV-syn-GCAMP6s viral vector. The recordings were done either in complete darkness or in front of a gray screen at low luminance (0.57 cd/m$^2$; when screen is on, the diffuse light allows the mouse to see its immediate surroundings). To monitor gaze stability, pupil position and diameter were tracked during the experiment (*Figure 2C,D*). V1 was identified using Fourier intrinsic imaging (*Marshel et al., 2011*) as the largest retinotopic field in the visual areas (*Figure 2E,F*), and all two-photon imaging fields-of-view were mapped to the retinotopic field using blood vessel landmarks (*Figure 2E*). All neurons imaged outside V1 (*Figure 2E,F*) were analyzed separately.

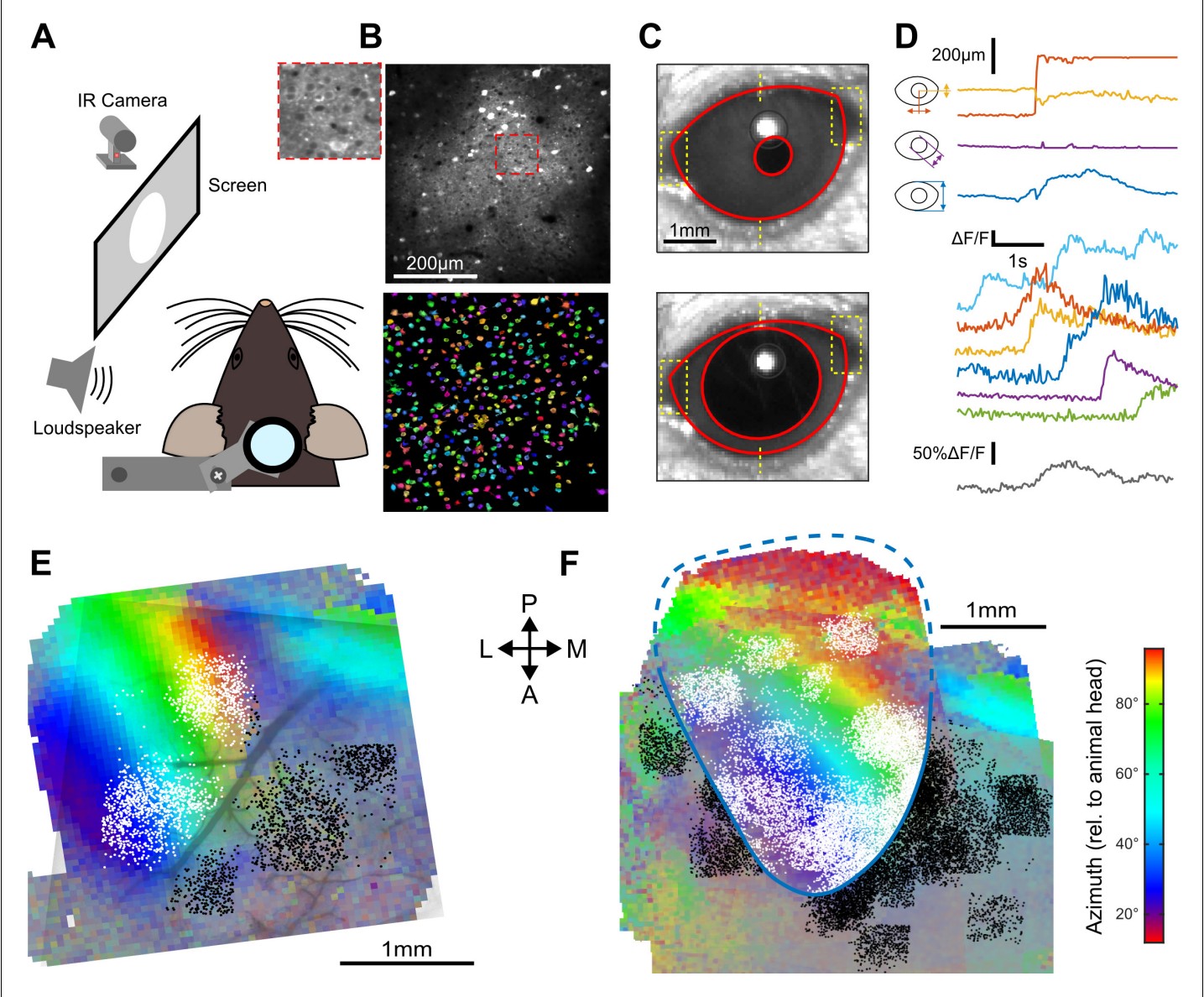

**Figure 2.** Retinotopically mapped two-photon imaging fields in V1. (**A**) Sketch of the experimental setup. (**B**) (top) Example of a 0.5 × 0.5 imaging field of view showing GCAMP6s-expressing neurons in V1 (top with magnification in inset) and the ROIs that were automatically detected as putative neurons (bottom, see Materials and methods). (**C**) Examples of eye tracking images. (**D**) (top) Eye tracking showing a large saccade during a blank trial. The blue, purple, and yellow-red traces indicate apertures between eye lids, pupil diameter and x-y motion of pupil center. (bottom) Examples of raw GCAMP6s traces from individual neurons recorded concomitantly in V1 (each color is a single neuron, frame rate: 31 Hz) and population average (gray) showing saccade-related neuronal activity. (**E**) Example of a retinotopic map obtained with Fourier intrinsic imaging (see Materials and methods) and segregation between neuron locations inside (white) and outside (black) V1. The color code indicates the azimuth in the visual field. (**F**) Registered maps across nine mice.

DOI: https://doi.org/10.7554/eLife.44006.006

As visual search behavior can be influenced by auditory stimuli (*Stein and Stanford, 2008*), we first measured whether sounds motivated eye-movements. We found that sounds occasionally triggered predominantly horizontal saccadic eye movements, especially sounds with loud onsets (*Figure 3A*). Sound-induced changes in pupil diameter were also observed (*Figure 3B*). In the lit condition, the saccades triggered responses in V1 (e.g. *Figure 2D*). Therefore, we excluded all trials with a saccade larger than mouse visual acuity (2° of visual angle) from our analyses (*Figure 3A*). After this correction, we pooled together the activity of 18925 V1 neurons (35 sessions, nine mice)

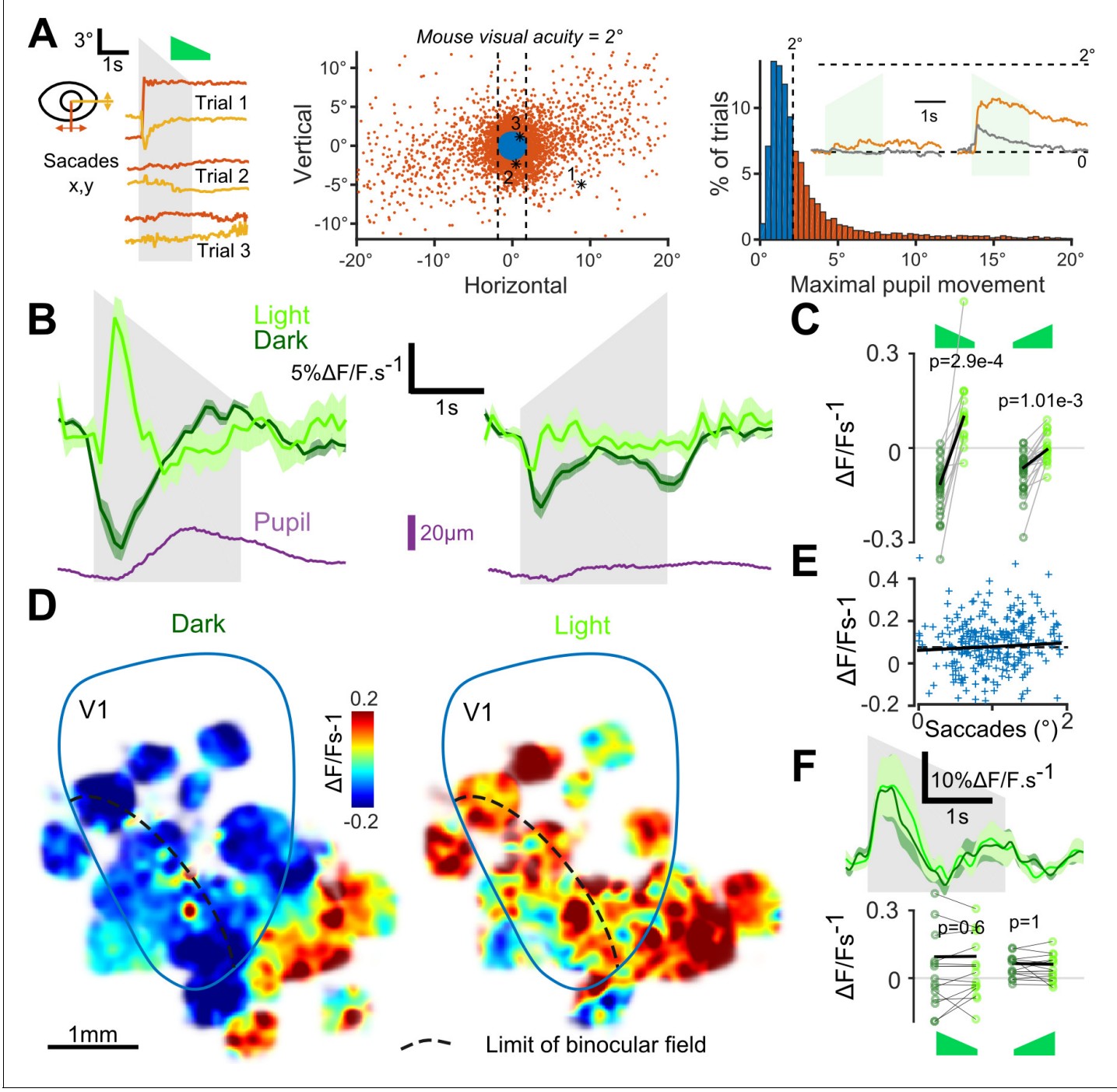

**Figure 3.** Auditory responses in V1 switch sign in the presence of visual inputs. (**A**) (left) Horizontal and vertical pupil movements recorded during three presentations of the down-ramp (gray shading). (center and right) Distribution of saccades across all mice (n = 7) and trials. The blue portion indicates trials where pupil movement did not exceed visual acuity in the mouse (2° of visual angle: 84% of all trials). The trials with larger eye movements (red) were removed from the analysis (trial filtering). Inset: average saccade responses to up-ramps and down-ramps, before (orange) and after (gray) trial filtering. (**B**) Averaged deconvolved calcium traces of V1 neurons in the light (light green) and in the dark (dark green) (6207 neurons, n = 17 sessions in seven mice). The purple line is the average pupil diameter. Left: Down ramp responses, Right: Up ramp responses. Up- and down ramping gray shadings indicate the timing of up- and down ramping sounds. (**C**) Average V1 responses are larger in the light than in the dark (n = 17 recording sessions in seven mice, Wilcoxon signed rank test). (**D**) Smoothed maps (Gaussian filter, σ = 320 μm) of the local responses to the down-ramping sound, averaged across sessions and animals after registration with respect to the retinotopic map. (**E**) Single trial saccade amplitudes (below the 2° visual acuity threshold) do not correlate with the amplitude of V1 population responses to sounds (Pearson correlation coefficient 0.05, p=0.42). (**F**) (top) Mean deconvolved signal for 2474 V1 projecting cells recorded in AC in light and in dark conditions. (bottom) Average onset response (0 to 500 ms) to up-

*Figure 3 continued on next page*

*Figure 3 continued*

and down-ramps for each recording session, showing no significant difference between light and dark conditions (Wilcoxon sign test, p=0.6 and p=1, n = 15 sessions).

DOI: https://doi.org/10.7554/eLife.44006.007

The following figure supplements are available for figure 3:

**Figure supplement 1.** Robust inhibition in response to sounds in V1.

DOI: https://doi.org/10.7554/eLife.44006.008

**Figure supplement 2.** Mean V1 responses (2226 neurons, n = 13 sessions in two additional mice) to a down-ramping sound (gray shading) in the dark, in the light and with the contralateral eye reversibly occluded as depicted in the sketch on the left.

DOI: https://doi.org/10.7554/eLife.44006.009

and observed that sounds alone trigger responses in supra-granular V1 neurons. The responses were stronger for the down-ramping sounds, similar to what we saw in the activity of V1-projecting AC neurons (*Figure 3B,C*). Strikingly however, we observed that the net population response was globally excitatory when mice were in the light and inhibitory when mice were in complete darkness (*Figure 3B,C*). The observed inhibition was seen consistently across cells and sound presentations, likely corresponding to a transient decrease in basal firing rates (*Figure 3—figure supplement 1*). Interestingly, covering the contralateral eye was sufficient to obtain the same inhibition as in darkness (*Figure 3—figure supplement 2*), suggesting a possible role of direct unilateral visual inputs in the context dependence of auditory responses in V1. As a sanity check, we verified that unfiltered micro-saccades did not cause the excitatory effects, and indeed the amplitude of these small saccades was not correlated with sound responses (*Figure 3E*). Also, excitatory responses to sounds in the light cannot be attributed to sound-induced pupil dilation as these movements occurred after V1 responses (*Figure 3B*). Taken together, our observations corroborate the idea that the sign of V1 auditory responses depends on the visual context (darkness vs dimly lit visual scene).

To test whether this context-dependence was a general property of auditory inputs to non-auditory cortical areas or if it was specific to auditory inputs impacting V1, we measured auditory responses in the secondary visual or associative areas next to V1 which also receive inputs from AC (*Zingg et al., 2014*). Unlike V1, these areas displayed excitatory responses in both the lit and dark conditions (*Figure 3D*). We then wondered whether the context-dependence was a property arising in the circuit of V1 or if it is due to a modulation of auditory inputs by the light context. We imaged sound responses in V1-projecting neurons of auditory cortex specifically labeled using the CAV-Cre retrograde virus approach (see *Figure 1*). We observed no response modulation between the dark and lit conditions (*Figure 3F*), indicating that context-dependent modulation arises in V1.

Finally, we verified that, as suggested by previous studies (*Ibrahim et al., 2016*; *Iurilli et al., 2012*), context-dependent auditory inputs to V1 are mediated by direct AC to V1 projections. Injections of the GABA-receptor agonist muscimol into AC during imaging almost completely abolished auditory responses in V1 and in the neighboring areas (*Figure 4* and *Figure 4—figure supplement 1*). We also used targeted chemogenetic inhibition (systemic CNO injections) of V1-projecting neurons in AC, expressing the hM4Di channel by co-injection of a CAV-Cre virus in V1 and an AAV8-flex-hM4di in AC (*Figure 4*). This produced a decrease of sound responses which was consistent across sound repetitions for the imaged neural population (*Figure 4C*). However, probably because this strategy impacts an incomplete subset of all V1-projecting neurons, the effects were smaller and more variable across experiments (*Figure 4—figure supplement 1*). Taking these results together, we conclude that V1 implements a mechanism that inverts the impact of AC inputs depending on whether or not visual information is available.

## Layer 1 contains an interneuron sub-population delivering context-dependent inhibition

To investigate the circuit mechanisms of this context-dependence, we reasoned that the occurrence of context-dependent inhibition in L2/3 neurons could result from a cortical inhibitory population that receives auditory inputs but is itself inhibited in light condition. To identify such a population, we used GAD2-Cre x flex-TdTomato mice injected with AAV-syn-GCAMP6s to image identified excitatory and inhibitory neurons (*Figure 5A*) in L1 and L2/3 where AC to V1 projections are concentrated (*Ibrahim et al., 2016*). We observed that in layer 2/3 both excitatory (n = 4348) and inhibitory

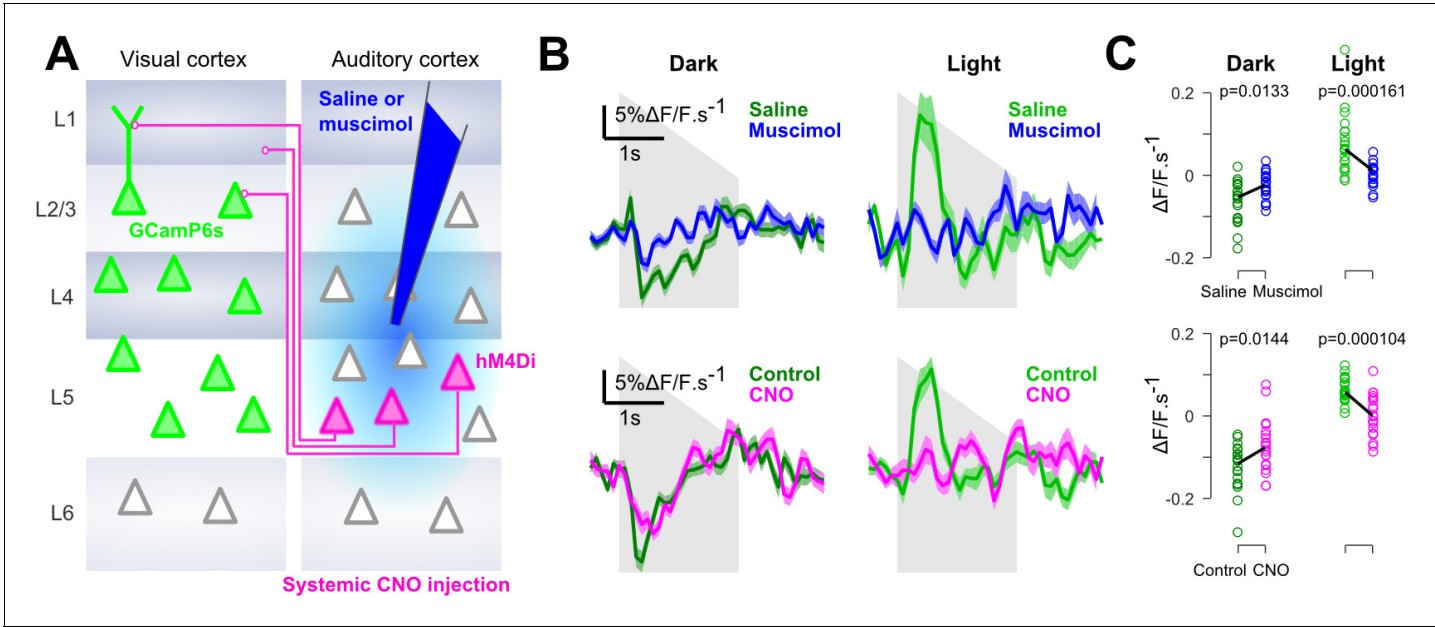

**Figure 4.** Auditory responses in V1 are caused by direct AC projections. (**A**) Schematics of the AC inactivation experiment (blue) and of the chemogenetic inactivation experiment in which V1-projecting neurons in AC are specifically silenced (magenta). (**B**) (top) Mean deconvolved calcium signal of V1 neurons in dark (dark green) or in light (light green) during saline injections (n = 765 neurons in four experiments and three mice) or muscimol inactivation of AC (n = 825 neurons in five experiments, same three mice). (bottom) Same as top graphs but for chemogenetic inactivation of V1-projecting neurons in AC (same n = 743 neurons before and after CNO activation, in three experiments and two mice). Shading indicates SEM. (**C**) Mean deconvolved calcium signal from the population responses shown in B for the saline (green) versus muscimol (blue) conditions and for Control (green) versus CNO (magenta) conditions (computed from 200 to 500 ms after sound onset for the negative responses in the dark, and from 200 to 1000 ms for the positive response in the light). The p-values are derived from a two-sided Wilcoxon sign test (n = 40 stimulus repetitions).
DOI: https://doi.org/10.7554/eLife.44006.010

The following figure supplement is available for figure 4:

**Figure supplement 1.** Distribution of AC inactivation effects across experiments for the data shown in *Figure 4B and C*.
DOI: https://doi.org/10.7554/eLife.44006.011

neurons (n = 726) were globally inhibited by sounds in the dark and excited in the light, while the layer 1 interneuron population seemed globally unaffected in the dark and excited in the light (*Figure 5B*).

However, the population trend concealed functionally distinct subpopulations. When we tested for significant positive or negative responses in the dark in single neurons (Wilcoxon rank-sum test, p<0.01), we found a large fraction of neurons significantly inhibited, but also, a subpopulation of 12.8% of all L1 inhibitory neurons that were significantly activated in the dark (*Figure 5C*). This positive response was rare in L2/3 with only 2.1% of inhibitory and 1% of excitatory cells significantly activated (chance levels = 1%, *Figure 5C*). When plotting the responses of L1 neurons in the dark against their responses in the light (*Figure 5D*), it is apparent that the L1 neurons that are excited by sounds in the dark tend to be less excited in the light (e.g. *Figure 5E*). In contrast, L1 neurons inhibited or unaffected by sounds in the dark became more activated in the light (*Figure 5D*). Consistently, statistical assessment showed that a significant percentage (5.3%) of L1 neurons (Wilcoxon rank-sum test, p<0.01) respond less in light than in dark (*Figure 5F*), and that almost all these neurons are significantly excited in the dark (4.9%, *Figure 5G*). In contrast, based on the same tests, there was no significant population of L2/3 neurons that responded less in the light than in the dark (*Figure 5F,G*). Thus, we conclude that layer 1 contains the only supragranular subpopulation of GABAergic neurons that can provide a context-dependent, sound-induced inhibition gated by visual inputs. These neurons are thus good candidates to mediate the context-dependence of sound responses observed in the bulk of L2/3 V1 neurons (*Figure 5H*).

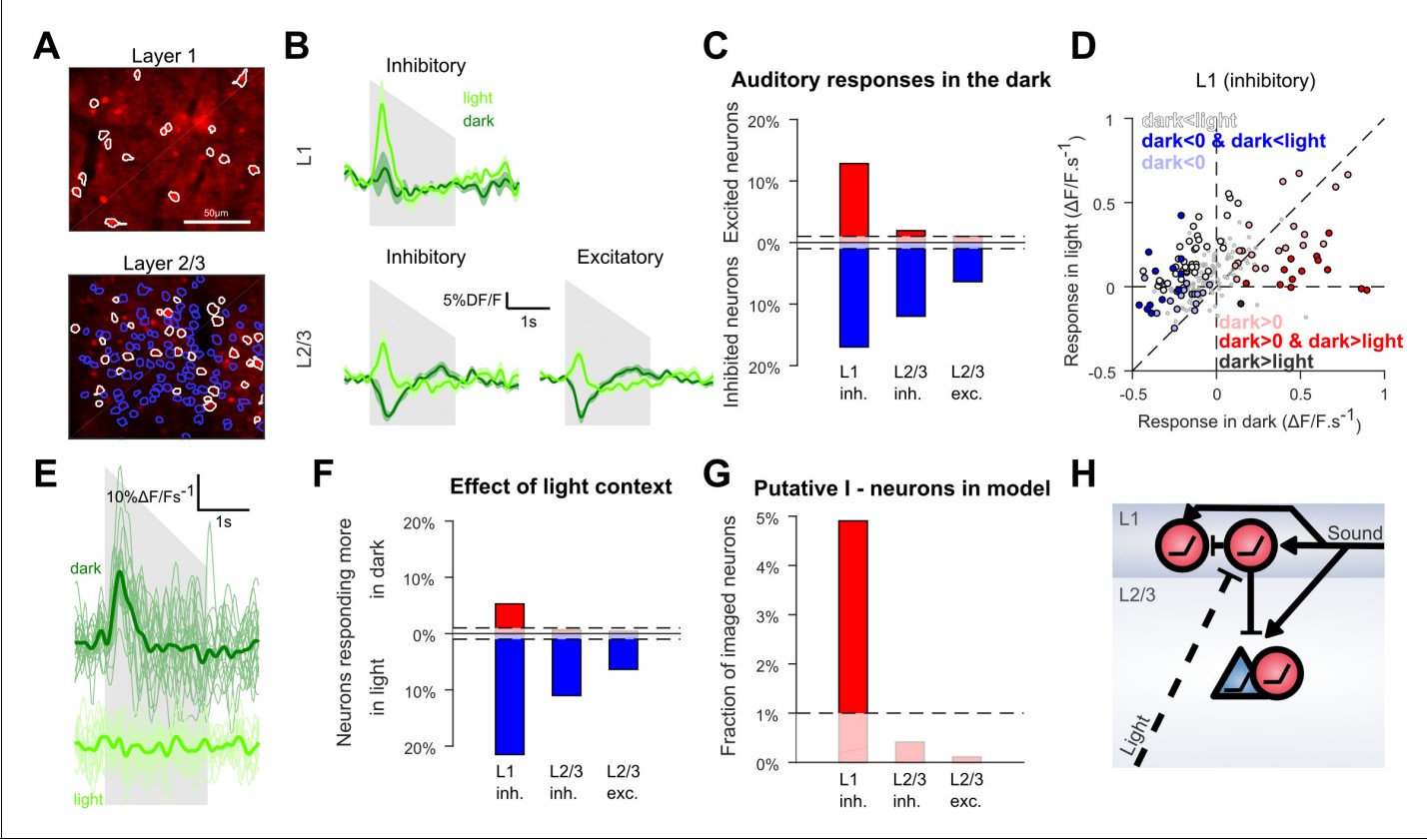

**Figure 5.** Context-dependent inhibition by sound is mediated by a sub-population of L1 interneurons. (**A**) In vivo two-photon images of GAD2-positive V1 neurons in L1 and L2/3 expressing td-Tomato. Superimposed are the contours of the active regions of interest identified by GCAMP6s imaging, blue: putative pyramidal cells, white: putative GABAergic cells. (**B**) Mean responses to the down-ramping sound (gray shading) for inhibitory and excitatory neurons in L2/3 and L1. (**C**) Fraction of neurons significantly excited (red) or inhibited (blue) by sounds for each layer and cell type (Wilcoxon rank sum test, p<0.01: the dashed lines indicate chance level). (**D**) Scatter plot of the mean responses of all L1 inhibitory neurons to the down-ramping sound in the light against in the dark. Small gray dots indicate neurons that do not significantly respond in any condition. Larger, colored dots indicate significantly responding neurons. Dark blue indicates neurons significantly inhibited in the dark and responding more in light than in dark. Dark red dots indicate neurons significantly activated in the dark and responding more in dark than in light. Neurons responding equally in dark and light are marked with light blue (negative response) and light red (positive response) dots. Neurons that are non-responsive in the dark but respond more or less in the light are marked in white and dark, respectively. (**E**) Single trial deconvolved traces for an L1 interneuron responding to a down-ramping sound (gray shading) in the dark and inactive in the light. (**F**) Fractions of neurons responding significantly more in light than in dark (blue) or more in dark than in light (red) (Wilcoxon rank sum test, p<0.01). (**G**) Fraction of neurons that can mediate context-dependent inhibition in each layer that is interneurons significantly excited by sounds in the dark and responding significantly less in the light than in the dark. (**H**) Simplified schematic of the proposed mechanism for context-dependent auditory responses in V1. Cells unaffected by visual context are not displayed.

DOI: https://doi.org/10.7554/eLife.44006.012

## Loud onsets boost the representation of coincident visual events

As we have shown that sounds can excite neurons in V1 in lit conditions, we next wanted to know the impact of AC inputs to V1 on the representations of coincident visual inputs. To this end, we recorded 9849 L2/3 neurons (23 sessions, seven mice) in mice injected with AAV-syn-GCAMP6s in V1. In these experiments, we presented the up-ramping and down-ramping auditory stimuli coincidently with a white looming or receding disk on the screen (sounds and visual stimuli were 2 s duration). The loudspeaker was placed such that auditory and visual stimuli came from the same direction. Unimodal stimuli were also delivered to assess additivity for the bimodal conditions.

We first observed that 11% of the V1 neurons displayed significant (Generalized Linear Model, p<0.05) supra-additive boosting of their visual responses when coincident with the loud onset of down-ramps (e.g. ***Figure 6A***), an effect even visible at the population level (***Figure 6B***) and dependent on AC as assessed with muscimol silencing (***Figure 6D,E***). Because looming and receding disks

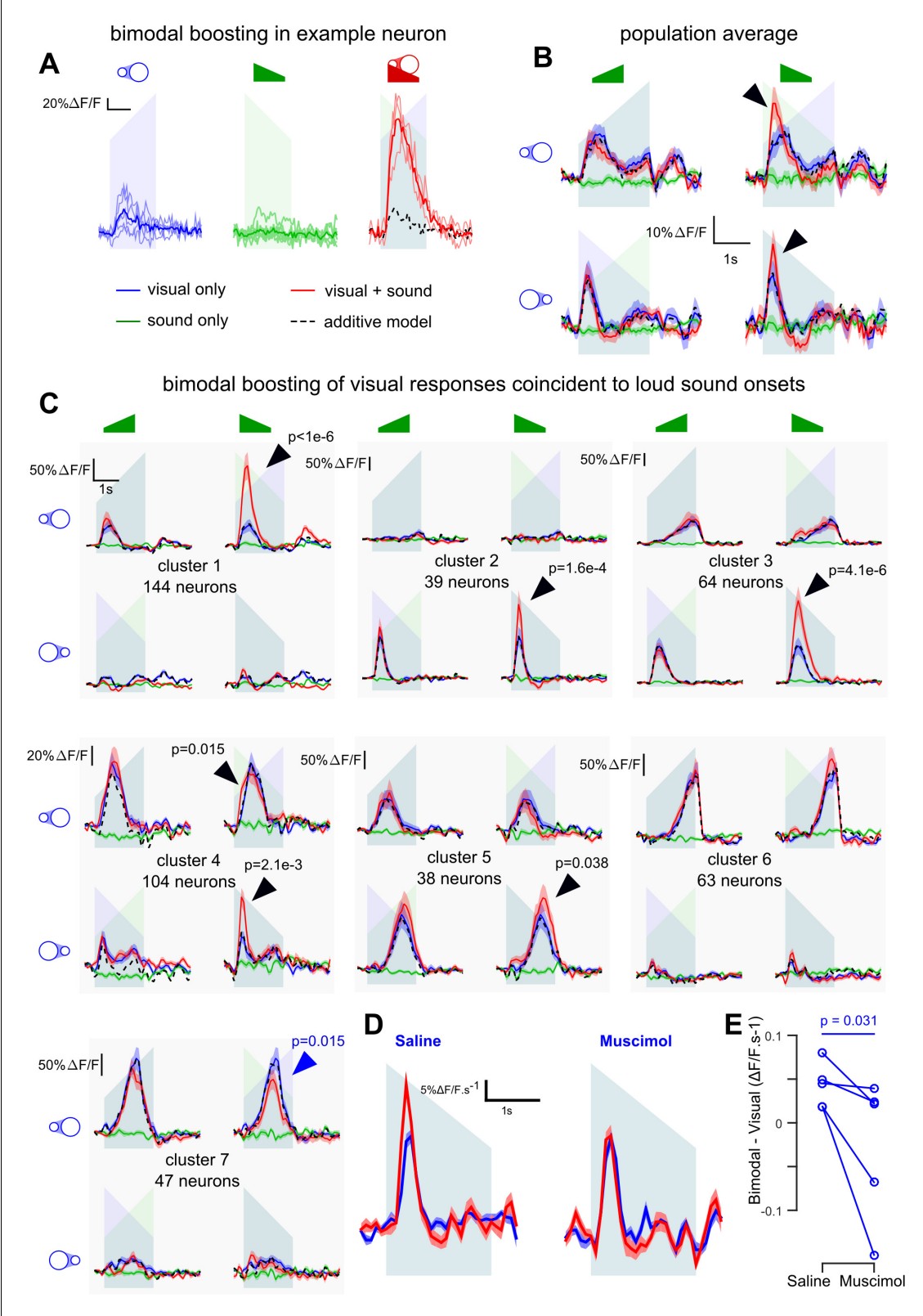

**Figure 6.** V1 neurons are boosted by the coincidence of loud auditory onsets with visual onsets. (**A**) Sample neuron in V1. Raw GCAMP6s traces for a visual stimulus (blue), a sound (green) and a combination of both (red) show a large neuronal response for the bimodal condition compared to the additive prediction (dashed black line) based on unimodal responses. Individual trials (thin lines) and their average (thick lines) show that the multisensory responses are robust for this neuron. (**B**) Average deconvolved calcium responses (mean ±sem) in V1 for up- and down-ramping sounds

*Figure 6 continued on next page*

*Figure 6 continued*

(green), looming and receding visual stimuli (blue) and the four bimodal combinations. Black dashed lines: linear prediction. (**C**) Uni- and bimodal responses, displayed as in B., of the seven different functional cell types identified by hierarchical clustering. (**D**) Average deconvolved responses to the receding disk in absence (blue) and presence (red) of a simultaneous down-ramping sound, upon saline injection in AC (left, n = 765 neurons in four experiments and three mice) and upon muscimol inactivation of AC (right, n = 825 neurons in five experiments, same three mice). (**E**) Distribution of AC inactivation effects across experiments for the data shown in D, one-sided Wilcoxon signed-rank test (n = 5). Throughout the figure the ramping green shadings mark the presentation of up- or down-ramping sounds, and the ramping blue shadings the presentation of looming and receding visual stimuli.

DOI: https://doi.org/10.7554/eLife.44006.013

The following figure supplement is available for figure 6:

**Figure supplement 1.** Bimodal responses in the area anteromedial to V1.

DOI: https://doi.org/10.7554/eLife.44006.014

trigger different types of responses in V1, we performed a clustering in order to identify the main types of visual responses and sound-induced modulations (see Materials and methods, note that only the V1 neurons with high signal-to-noise and sufficient number of trials without eye movement, n = 499, were clustered). This analysis identified seven distinct clusters (*Figure 6C*) capturing 49.5 ± 11.2% of the response variance. Four clusters (#1,4,6,7; 358 neurons) responded more to looming than receding disks, and three clusters (#2,3,5; 141 neurons) had the opposite preference. Four clusters (#1,2,6,7) were tightly direction-specific and responded only for their preferred stimulus, at its beginning or end, while the three others (#3,4,5) were less specific to direction, but also responded to specific parts of the stimulus (clusters 3 and 4: larger disk; cluster 5: smaller disk). This specificity was probably due to the location in the retinotopic field of the neurons with respect to the stimulus center.

The bimodal conditions revealed pronounced supra-additive responses. Down-ramping sounds boosted the responses of all cell clusters that responded to the onset of visual stimuli. (*Figure 6C*, black arrows for clusters 1–5, bootstrap test, see Materials and methods). Very little boosting was observed with the quiet onset of up-ramps (*Figure 6C*), consistent with its reduced representation in AC to V1 projections (*Figure 1*). Responses occurring towards the end of the stimulus were minimally (although significantly, cluster 5) boosted, probably because they correspond to auditory features ('loud and quiet tonic') that have less of an impact on V1 (see *Figure 3B*). Our data suggest that visual responses in V1 are boosted preferentially by the coincidence of loud sound onsets with respect to other basic envelope features. Note that a moderate response suppression was also observed, but only in cluster 7 (*Figure 6C*, blue arrow). Moreover, the boosting of visual responses appeared to be a strong feature of V1, as the same analysis identified only one supra-linear cluster out of six clusters in the associative area medial to V1 (*Figure 6—figure supplement 1*).

## Interplay between AC inputs and neuronal non-linearities can explain all auditory-driven responses in V1

How can the same circuit implement context-dependent sound-driven inhibition and a boosting of specific visual responses? To propose a mechanism, we established a minimal model in which AC projections excite both a population of excitatory neurons (pyramidal cells) and a population of interneurons in V1 (putatively, the identified L1 population), consistent with connectivity studies (*Ibrahim et al., 2016*). The interneuron population directly inhibits excitatory neurons in our model (*Figure 7A*). We used two critical mechanisms to render interneurons context-dependent: (i) a connection which inhibits interneurons in the presence of visual inputs (light modulation, *Figure 7A*, and (ii) a non-linear response function (response threshold). Then, provided that they are close to activation threshold (i.e. spike threshold) in the dark and that inhibition by visual inputs brings them well below threshold in the lit condition, interneurons will be less active in the light than in the dark. As a result, the excitatory population experiences a dominance of sound-driven excitation in the light and of sound-driven inhibition in the dark (*Figure 7B,C*) as observed experimentally. Biophysically, this mechanism requires a tonic inhibitory drive to the identified subset of L1 interneurons which could be provided by another inhibitory population. Interestingly, we observed that the subpopulation of L1 interneurons that are less responsive to sounds in light also have a smaller baseline fluorescence

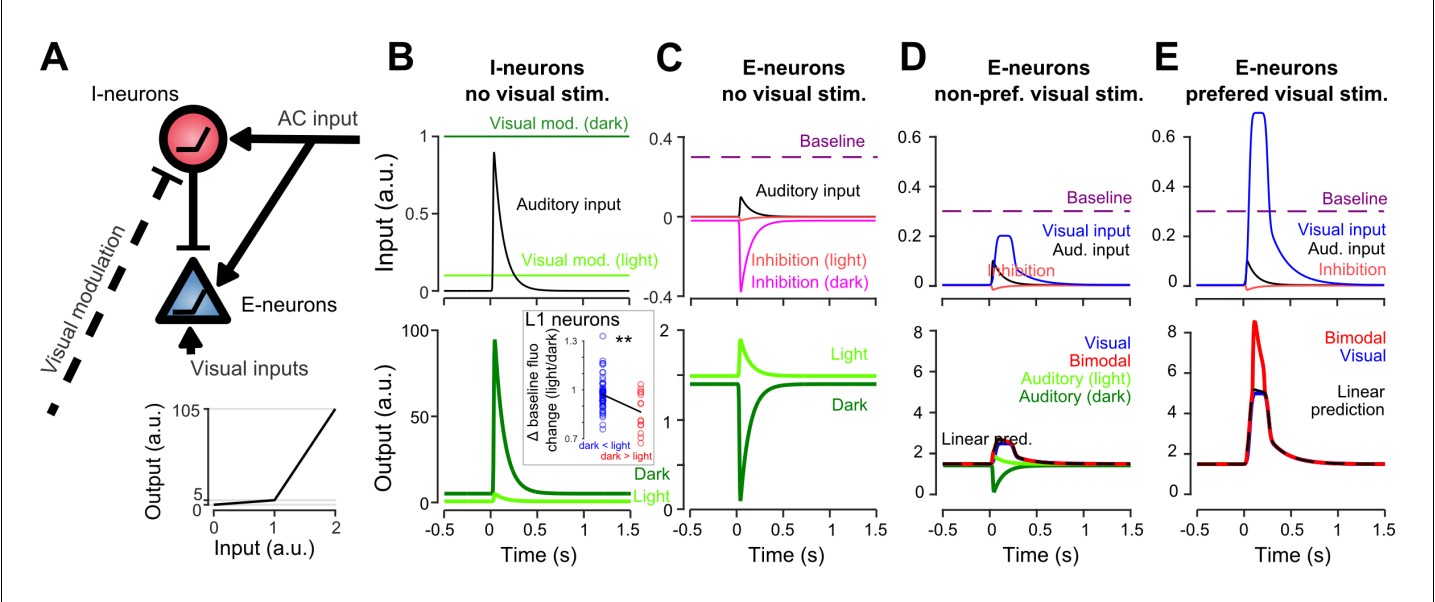

**Figure 7.** Non-linear inhibitory and excitatory neurons reproduce gated-inhibition and bimodal boosting. (A) Sketch of a minimal model for the switch between negative and positive sound responses in dark vs in light and boost of visual responses to sounds. An inhibitory (I-neurons) population endowed with a non-linear response function (bottom) receives a positive auditory input and a negative visual input. When both inputs are active (lit condition), the auditory drive to I-neurons is below threshold and no inhibition is delivered to the excitatory non-linear neuronal population (E-neurons). In the dark condition, the visual modulation is reduced releasing sound triggered inhibition. Both I- and E-neurons are non-linear with a low and a high output gain. Auditory and visual inputs are typically converted with a low-gain, while coincident auditory and preferred-visual inputs pass the threshold and yield a high-gain output (boosting). (B) Input currents and output response of I-neurons during auditory simulation in light and dark conditions. Inset: Change in baseline fluorescence between light and dark conditions in L1 interneurons. A significant baseline decrease in light is observed for interneurons that are more sound responsive in dark than in light (i.e. putative I-neurons, red) as compared to interneurons that are more responsive in the light (Wilcoxon rank sum test, p=0.00989, n1 = 57 'dark <light' cells, n2 = 14 'light <dark' cells). (C) Same for E-neurons. (D) Input currents and output response of E-neurons during visual simulation alone (blue) and bimodal stimulation for a sub-threshold (non-preferred) visual input. Superimposed are the auditory responses in light and dark from panel (C) (E) Same as in D. but for a preferred, supra-threshold visual input.
DOI: https://doi.org/10.7554/eLife.44006.015

in light, an effect that is not observed for other L1 interneurons (*Figure 7B*). This could be a possible manifestation of the tonic drive hypothesized in our model.

We then showed that the same model can reproduce boosting of visual responses by sounds (*Figure 7D,E*) if excitatory neurons also have a simple non-linear response function. When excitatory neurons have a low, subthreshold output gain and a high, supra-threshold gain (*Figure 7A*), it is possible to reproduce the observations that the response to sound alone (low gain regime) is weaker than the bimodal boosting effect (high-gain regime) (*Figure 7D–E*). With this design, we also reproduced the observation that boosting mainly occurs for the preferred visual stimulus (*Figures 6C* and *7D–E*), provided that the auditory and non-preferred visual inputs are driving excitatory neurons in the low-gain regime (*Figure 7D*) while the preferred input brings excitatory neurons close to or above the high-gain threshold (*Figure 7E*). In this case, the auditory response sums with a high gain for the preferred stimulus and a low gain for the non-preferred. While less conventional than a regular spike threshold, this mechanism could be biophysically implemented with non-linear dendritic processing. Apical dendrites of layer 2/3 pyramidal cells are known to produce large calcium spikes which boost somatic output when somatic activity coincides with excitatory inputs arriving in the apical tree (*Larkum et al., 2007*). As AC axon terminals are concentrated in layer 1 where they likely contact the apical dendrites of pyramidal cells, sound-driven boosting of visual responses could rely on this dendritic phenomenon.

Taken together, our data and modeling indicate that a minimal circuit including a direct excitation from auditory cortex onto L2/3 pyramidal cells and an indirect inhibition via a subpopulation of L1 interneurons is sufficient to provide a context-dependent auditory modulation that emphasizes visual events coincident with loud sound onsets.

# Discussion

Using two-photon calcium imaging in identified excitatory and inhibitory neurons during auditory and visual stimulation, we demonstrated three important features of AC to V1 connections. First, we showed that V1-projecting neurons have a different bias in the distribution of sound envelope features they encode, as compared to layer 2/3 neurons in AC and also, although to a lesser extent, as compared to layer 5 in which they are located. In particular, the V1-projecting neuron population predominantly encodes loud sound onsets. Second, they are gated by a context-dependent inhibitory mechanism likely implemented by a subpopulation of L1 interneurons. Third, their impact on V1 representations is to produce a strong boosting of responses to dynamical visual stimuli coincident with different sound envelope features, particularly with loud sound onsets.

The shifts in the distribution of sound envelope features that we observed across layers and cell subpopulations is an interesting case of coding bias in different neural populations of auditory cortex (*Figure 1*). As observed earlier, in layer 2/3, there is a balance between the main neuron types that signal the beginning and the end of sounds with equivalent fractions of neurons responding to quiet or loud onsets and loud offsets (*Figure 1F*, note that 'tonic +loud OFF' neurons are included in this count. Quiet offset neurons are rare and not robustly detected across datasets. Detection of tonic responses by clustering is also variable.). In upper layer 5, the distribution becomes more biased toward loud onsets at the expense of offset neurons, a trend that is further increased in the layer 5-located V1-projecting cells in which half of the neurons respond to loud onsets and 10% of the cells to loud offsets. Similar results have been found in somatosensory cortex for neurons projecting to secondary somatosensory or motor cortex (*Chen et al., 2013a*). In AC, it is well-established that specific acoustic features are related to the localization of neurons tangentially to the cortical surface in different subfields (*Kanold et al., 2014*; *Schreiner, 1995*; *Schreiner and Winer, 2007*) and that, at least in higher mammals such as cats, particular auditory cortical fields may play different functional roles in behavior, such as sound localization (where) or identification (what) (*Lomber and Malhotra, 2008*). Our results show that projection specificity can also be a determinant of feature selectivity, at least for some intensity modulation features at sound onsets and offsets.

As a result of these fine adjustments of feature distributions, layer 2/3 neurons in V1 are more impacted by abrupt sounds with loud onsets (e.g. down-ramps) than by sounds with softer onsets (e.g. up-ramps). This asymmetry is particularly striking for excitatory effects seen in light or with visual stimuli (*Figures 3B* and *6*), but less pronounced for inhibitory effects in the dark (*Figure 3B*). It is thus possible that the projection targets also depend on encoded features. For example, L2/3 V1 pyramidal cells may receive projections almost exclusively from AC loud onset neurons while interneurons providing inhibition in the dark still receive a significant fraction of quiet onset and loud offset inputs.

The context-dependence of sound-triggered inhibitory and excitatory auditory responses in V1 is interesting in two respects. First, it reconciles conflicting reports of sound-induced inhibition or excitation in V1 (*Ibrahim et al., 2016*; *Iurilli et al., 2012*). Second, context-dependent inhibition could serve to decrease V1 activity in the dark when visual information is irrelevant for sound source localization, while leaving possibilities to integrate auditory and visual information in light conditions. The context-dependence of auditory responses in V1 is reminiscent of the recent observation that AC to V1 projections, activated by loud sounds and acting through L1, boost preferred orientation responses in V1 and inhibit non-preferred orientations (*Ibrahim et al., 2016*). The two effects share several properties including selectivity to high sound levels, the dominance of inhibition for weaker visual inputs and of excitation for stronger visual inputs, and the involvement of L1 neurons for mediating cross-modal inhibition. The main discrepancy is the apparent lack of visual input specificity of our illumination-dependent excitation of V1 neurons by AC inputs alone (i.e. no inhibition of responses to non-preferred visual stimuli). This could be due to differences in the type of visual inputs used. One important result appearing in our data is that only a fraction of all L1 interneurons have response properties compatible with the function of releasing a context-dependent, sound-triggered inhibition while these properties are not found in L2/3 neurons (*Figure 5C,F,G*). Thus, the silencing of visual processing by sounds in the dark is not a generic function of L1 interneurons, compatible with the observation that only VIP-negative interneurons receive AC inputs in V1 (*Ibrahim et al., 2016*) and the observation that L1 interneurons serve other functions (*Hattori et al., 2017*), such as disinhibition of L2/3 in various contexts (*Fu et al., 2014*; *Lee et al., 2013*;

*Letzkus et al., 2015*; *Pfeffer et al., 2013*; *Pi et al., 2013*). An interesting question that cannot be addressed with the current data is how the specific L1 interneurons that mediate sound-induced inhibition are themselves inhibited in the presence of visual inputs. Several putative mechanisms are compatible with the data, including disinhibition. We can however exclude that visual responses in AC (*Morrill and Hasenstaub, 2018*) modulate a fraction of the AC input to V1, based on context-independent responses of retrogradely identified V1-projecting cells (*Figure 3F*).

Interestingly, our data obtained outside V1 indicate that, independent of the exact mechanism of context-dependence of auditory responses, this phenomenon is restricted to V1 (*Figure 3D*). This indicates that AC projections to associative areas impact the local circuit differently than in V1. A possible explanation could be that projections to excitatory neurons are favored or that the layer 1 interneurons that provide strong sound-driven inhibition in V1 in the dark are nonexistent or excluded by AC projections in associative areas.

Another important phenomenon observed in our data is the boosting of visual responses coincident with loud sound onsets (*Figure 6*), an effect that could even potentially be amplified if visual stimuli led sounds by a few tens of milliseconds as observed in monkey auditory cortex (*Kayser et al., 2008*). Our model suggests that this can result directly from the non-linear summation of visual and auditory inputs in V1 pyramidal cells, while sound-triggered inhibition is abolished by the visual context (*Figure 7*). The large amplitude of the sound-induced boosting as compared to the weaker responses observed when sounds are played without coincident visual input (*Figures 3* and *5* vs *Figure 6*, see also *Figure 7*) points toward a threshold mechanism, allowing for amplification of the auditory input only if sufficient visual input concomitantly arrives. Such a gated amplification mechanism could be implemented directly in L2/3 pyramidal neurons using the direct inputs they receive from auditory cortex (*Ibrahim et al., 2016*; *Leinweber et al., 2017*). For example, amplification of L1 input by coincident somatic inputs triggering calcium spikes in apical dendrites has been described both in L5 (*Larkum et al., 1999*) and L2/3 (*Larkum et al., 2007*) pyramidal cells. While this is a good candidate cellular mechanism for the observed boosting effect, we cannot exclude alternative or complementary mechanisms such as a disinhibition mediated by a subpopulation of L1 interneurons (*Ibrahim et al., 2016*; *Lee et al., 2013*; *Pfeffer et al., 2013*; *Pi et al., 2013*), a different population from the subpopulation providing direct inhibition (*Figure 5*). Also, as down-ramping sounds are startling stimuli, a cholinergic input to visual cortex, known to increase cortical responses via disinhibition (*Fu et al., 2014*; *Letzkus et al., 2015*; *Letzkus et al., 2011*) could also contribute to modulation of visual responses by sounds in V1. Independent of the mechanism, our data show that sound-induced boosting is a strong effect in V1 which tightly relates to coincidence with the loud onsets of sounds. An interesting hypothesis for the role of this mechanism, among other possibilities, is that this effect could provide a powerful way to highlight, in visual space, the visual events potentially responsible for the sound, and thus might be an element of the cortical computations related to the localization of sound sources (*Stein and Stanford, 2008*).

## Materials and methods

### Animals

All animals used were 8–16 week-old male C57Bl6J and GAD2-Cre (Jax #010802) x RCL-TdT (Jax #007909) mice. All animal procedures were approved by the French Ethical Committee (authorization 00275.01). Animals were allocated in experimental groups without randomization and masking, but all scoring and analysis is done with computer programs.

### Two-photon calcium imaging in awake mice

Three to four weeks before imaging, mice were anaesthetized under ketamine medetomidine. A large craniotomy (5 mm diameter) was performed above the right primary visual cortex (V1) or the right auditory cortex (rostro-caudal, lateral and dorso-ventral stereotaxic coordinates from bregma, V1: −3.5, 2.5, 1 mm; AC: −2.5, 4, 2.2 mm). For the imaging of the auditory cortex (AC), the right masseter was removed before the craniotomy. We typically performed three to four injections of 150 nl (30 nl.min$^{-1}$), AAV1.Syn.GCaMP6s.WPRE virus obtained from Vector Core (Philadelphia, PA) and diluted x10. For imaging V1-projecting neurons in AC, a stereotaxic injection of 5x diluted CAV-2-Cre viral solution (obtained from the Plateforme de Vectorologie de Montpellier) was performed

just before the AC craniotomy. The craniotomy was sealed with a glass window and a metal post was implanted using cyanolite glue followed by dental cement. A few days before imaging, mice were trained to stand still, head-fixed under the microscope for 30–60 min/day. Then mice were imaged 1–2 hr/day. Imaging was performed using a two-photon microscope (Femtonics, Budapest, Hungary) equipped with an 8 kHz resonant scanner combined with a pulsed laser (MaiTai-DS, SpectraPhysics, Santa Clara, CA) tuned at 920 nm. We used 20x or 10x Olympus objectives (XLUMPLFLN and XLPLN10XSVMP), obtaining a field of view of 500 × 500, or 1000 × 1000 microns, respectively. To prevent light from the visual stimulation monitor from entering the microscope objective, we designed a ring silicon mask that covered the gap between the metal chamber and the objective. Images were acquired at 31.5 Hz during blocks of 5 s interleaved with 3 s intervals. A single stimulus was played in each block and stimulus order was randomized. All stimuli were repeated 20 times except drifting gratings (10 repetitions).

Drifting square gratings (2 Hz, 0.025 cyc/°) of eight different directions (0° from bottom to top, anti-clockwise, 45°, 90°, 135°, 180°, 225°, 270°, 315°), and a size- increasing (18° to 105° of visual angle) or decreasing white disk over a black background were presented on a screen placed 11 cm to the mouse left eye. The luminance of the black background and white disks was 0.57 and 200 cd/m$^2$, respectively. The screen (10VG BeeTronics, 22 × 13 cm) was located 11 cm from the contra-lateral eye, thus covering 90° x 61° of its visual field including some of the binocular segment. For auditory cortex recording, we used 250 ms constant white noise sounds at four different intensity modulations, and two up- and down-intensity ramping sounds between 50 and 85 dB SPL. For all sounds, the intensity envelope was linearly ramped from or to 0 during 10 ms at the beginning and the end of the sound. Note that microscope scanners emit a constant background sound of about 45 dB SPL. For visual cortex experiments, we used only the two intensity ramps and added two frequency modulated sounds, going linearly from 8 kHz to 16 kHz and vice versa. Up- and down- amplitude ramps were combined with the increasing and decreasing disks to form four multisensory stimuli. The loudspeaker was placed just next to the stimulation screen, facing the contralateral eye. All stimuli were 2 s long. Auditory and visual stimuli were driven by Elphy (G. Sadoc, UNIC, France). All sounds were delivered at 192 kHz with a NI-PCI-6221 card (National Instruments) an amplifier and high-frequency loudspeakers (SA1 and MF1-S, Tucker-Davis Technologies, Alachua, FL). Sounds were calibrated in intensity at the location of the mouse ear using a probe microphone (Bruel and Kjaer).

For V1 recordings, auditory only stimulations were performed in two different contexts: either in complete darkness (screen turned off in the sound and light isolated box enclosing the microscope) or in dim light (screen turned on with black background, luminance measured to 0.57 candela per square meter).

## AC inactivation experiments

To perform muscimol inactivation of AC during V1 imaging a stereotaxic muscimol injection (100 or 150 nL, 1 mg/mL) was performed before imaging under light isoflurane (~1.5%) anesthesia via a hole drilled through cement and bone on the lateral side of the V1 cranial window. The hole was sealed with Kwik-Cast and the animal was allowed to recover 20 min from anesthesia in its cage before being placed under the microscope and imaged during auditory-visual stimulation. Saline controls were performed using the same procedure on the next day, imaging the same V1 region and a similar optical plane (the exact same neurons were not tracked across days but the imaging plane was as similar as possible). To perform chemogenetic inactivation of V1 projecting neurons in AC during V1 imaging, three stereotaxic injections of undiluted AAV8-hSyn-DIO-hM4D(Gi)-mCherry virus were done in AC and immediately after CAV-2-Cre virus (5x) was co-injected with the AAV1.Syn. GCaMP6s.WPRE virus in V1 during window implantation. Four to five weeks later, a control imaging session was performed, followed by a subcutaneous CNO injection (3 mg/kg) under light isoflurane anesthesia (~1.5%) in the imaging setup. After a 60 min waiting period in the setup, the animal was imaged again to monitor the effect of the chemogenetic manipulation in the exact same neurons as during the control session.

## Intrinsic optical imaging recordings

To localize the calcium-imaging recordings with respect to the global functional organization of the cortex, we performed intrinsic optical imaging experiments under isoflurane anesthesia (1%). The brain and blood vessels were illuminated through the cranial window by a red (intrinsic signal: wavelength 780 nm) or a green (blood vessel pattern: wavelength 525 nm) light-emitting diode.

To localize the visual cortex, the reflected light was collected at 15 Hz by a charge-coupled device (CCD) camera (Foculus IEEE 1394) coupled to the epifluorescence light path of the Femtonics microscope (no emission or excitation filter). A slow drifting bar protocol was used: a white vertical bar was drifting horizontally over the screen width for 10 cycles at 0.1 Hz, from left to right in half of the trials, and from right to left in the other half. After band-passing, the measured signals around 0.1 Hz, and determining their phase in each pixel and for each condition, both the preferred bar location and the hemodynamic delay could be determined for each pixel, yielding azimuth maps as in (*Figure 2E*). Similarly, elevation maps were obtained in a subset of the animals using a horizontal bar drifting vertically. These maps coincided with those obtained in previous studies (*Marshel et al., 2011*), so we could determine V1 border as the limit between pixels displaying and not displaying retinotopy.

To localize the auditory cortex, the reflected light was collected at 20 Hz by a CCD camera (Smartek Vision, GC651M) attached to a custom-made macroscope. The focal plane was placed 400 μm below superficial blood vessels. A custom-made Matlab program controlled image acquisition and sound delivery. Sounds were trains of 20 white noise bursts or pure tone pips separated by smooth 20 ms gaps (down and up 10 ms linear ramps). We acquired a baseline and a response image (164 × 123 pixels,~3.7×2.8 mm) corresponding to the average images recorded 3 s before and 3 s after sound onset, respectively. The change in light reflectance (ΔR/R) was computed then averaged over the 20 trials for each sound frequency (4, 8, 16, 32 kHz, whitenoise). A 2D Gaussian filter (σ = 45.6 μm) was used to build the response map. The functional localization of the auditory cortex in this study corresponded to the response map produced by white noise.

## Eye tracking and trial filtering

Left eye measurements (eye size, pupil diameter, pupil movement) were made by tracking the eye using a CCD camera (Smartek Vision, GC651M). A Python software was used to capture images from the camera at 50 Hz, synchronized with the cortical recordings.

These movies were analyzed off-line using custom automatic Matlab programs that traced the contours, first of the eyelid, second of the pupil. The eye lid shape was approximated by two arcs and involved the estimation of 6 parameters (four for the coordinates of the two points where the arcs join, two for the y-coordinates of the crossings of the arcs with the vertical line at halfway between these two points; see *Figure 2C*, bounds for the parameters were set by hands and appear in yellow). The pupil shape was approximated by an ellipse, described by four parameters (center x and y, radius and eccentricity). Both estimations were performed by maximizing the difference between average luminance inside and outside the shape, as well as the luminance gradient normal to the shape boundary; in addition, they were inspected manually and corrected occasionally inside a dedicated graphical user interface. In the 'dim light' and 'bimodal' (but not the 'dark') contexts, it was observed that saccades correlated with population activity increases (e.g. *Figure 2D*), therefore we discarded all trials from these contexts displaying saccades. To do so, we computed the maximal distance between pupil center positions $(x_c, y_c)$ at different instants $t$ and $t'$ during the 2 s of stimulation $\max_{0<t,t'<2s} \sqrt{(x_c(t') - x_c(t))^2 + (y_c(t') - y_c(t))^2}$ and discarded trials where the change in gaze exceeded the mouse visual acuity of 2°, that is where this distance exceeded 57.6 μm (assuming an eye radius of 1.65 mm).

## Calcium imaging data summary

The data obtained from the different imaging experiments consisted of the following.

A1 recordings of auditory responses: 4616 cortical neurons from 29 L2/3 imaging areas in 11 animals. In 18 sessions (seven mice), animals were also stimulated with looming visual stimuli and bimodal stimuli from which we observed no response or modulation. 7757 neurons from upper layer 5 recorded in 12 sessions across four mice during sound only protocols. 2474 neurons projecting to

V1 recorded in 15 sessions across three mice during sound only protocols repeated in light or in dark conditions.

V1 recordings: 18,925 neurons from 35 L2/3 imaging areas in nine mice, and included the three 'dark', 'dim light' and 'bimodal' blocks in random order. Eye tracking was performed in 23 of these sessions (9849 neurons, seven mice; only these sessions where used for the analyses of the 'dim light' and 'bimodal' context, which required filtering out trials with saccades).

V1 recordings with labeling of Gad-positive interneurons: an additional 4348 neurons from 12 L2/3 imaging sessions (depth ranging from 140 to 300 microns), and 265 GAD2 positive neurons from 7 L1 imaging sessions (depth from 30 to 100 microns). All of these sessions had eye tracking, and included the 'dark' and 'dim light' blocks. 6 L2/3 sessions also included 'bimodal' blocks, whereas the 13 remaining sessions included a second 'dim light' block where the left (contralateral) eye was covered by a dark mask (*Figure 3F*).

## Calcium imaging pre-processing

Data analysis was performed with custom-made Matlab scripts available upon request. Every frame recorded was corrected for horizontal motion to a template image using rigid body registration (all sessions with visible z motion were discarded). Regions of interest were then automatically selected and human checked as the cell bodies of neurons with visually identifiable activity (*Roland et al., 2017*) and the mean fluorescence signal F(t) was extracted for each region. We also estimated the local neuropil signal $F_{np}(t)$ for each neuron and subtracted a fraction of it from the neuron signal $F_c(t)$ =F(t) - 0.7 $F_{np}(t)$. This fraction was set to 0.7, according to a previous calibration for GCAMP6s in mouse visual cortex (*Chen et al., 2013b*). Baseline fluorescence $F_{c0}$ was calculated as the minimum of a Gaussian-filtered trace over the 5 neighboring 5 s imaging blocks and fluorescence variations were computed as f(t) = ($F_c(t)$ - $F_{c0}$)/$F_{c0}$. Analyses were performed either on these normalized fluorescence signals, or on estimations of the firing rate obtained by temporal deconvolution (*Bathellier et al., 2012*; *Yaksi and Friedrich, 2006*) as r(t)=f'(t) + f(t)/τ, in which f'(t) is the first derivative (computed at successive samples separated by ~30 ms) of f(t) and τ = 2 s, as estimated from the decays of the GCAMP6s fluorescent transients (*Chen et al., 2013b*). This simple method efficiently corrects the strong discrepancy between fluorescence and firing rate time courses due to the slow decay of spike-triggered calcium rises (*Bathellier et al., 2012*), and was preferred to more advanced deconvolution methods (*Deneux et al., 2016b*) because it does not bias deconvolution towards absence of activity (i.e. interpreting small signal as noise) in cells displaying poor signal to noise ratio. However, it does not correct for the relatively slow rise time of GCAMP6s, producing a time delay on the order of 70 ms between peak firing rate and peak deconvolved signal.

## Calcium imaging analysis

Data analysis was performed with custom-made Matlab and Python scripts. Clustering of auditory responses in the auditory and visual cortices (*Figure 1*), and of bimodal conditions in the visual cortex (*Figure 6*) was performed using the following procedure. Deconvolved calcium responses were averaged across all valid trials (eye movement filtering). We then subtracted the baseline (average activity from 0 to 0.5 s before stimulus onset) and the average activity profile in trials with no stimulation. Hierarchical clustering was performed using the Euclidean metric and Ward method for computing distance between clusters. To determine the number of clusters, we moved down the clustering threshold (100 clusters for all AC datasets, 25 clusters for V1 data) until clusters became redundant (overclustering) as assessed visually. This method clusters neurons irrespective of whether they significantly responded to the stimuli. As a large number of neurons were not (or very weakly) responsive to the stimuli, a fraction of the clusters were merged into a cluster of 'noisy' non- or weakly responsive cells, which were eliminated from the final cluster list based on their heterogeneity measured as the mean correlation across the responses of each cell (only clusters with homogeneity larger than 0.4 were kept). These thresholds were chosen by visual inspection of the obtained clusters. Some clusters which were homogeneous but obviously captured a systematic perturbation of the signal (correlated noise) were also manually put in the non-responsive cluster. To make sure cell type distributions were not skewed by the fact that clustering outputs the most robust auditory responses (*Figure 1*), we re-aggregated neurons discarded as non-responsive if the mean correlation of their activity signature with any of the identified clusters was larger than 0.4. This procedure did

not change the mean cluster response profiles and did not qualitatively impact the conclusions drawn on the distributions of the different clusters in V1-projecting and L2/3 neurons.

## Awake electrophysiology

Multi-electrode silicon probe recordings were done in mice already implanted with a cranial window above AC at least 2 or 3 weeks before the experiment. The brain was exposed by removing some of the dental cement and part of the cranial window's cover slip with the animal under light isoflurane anesthesia (similar to intrinsic imaging). The brain was covered with Kwik-CastTM silicon (World precision Instruments) and the animal was removed from the setup to recover from anesthesia for at least 1 hr. Electrophysiological recordings were done using four shank Buzsaki32 silicon probes (Neuronexus). The animal was placed back on the setup-up and the Kwik-Cast was removed. The brain was then covered with warm Ringer's solution. The insertion of the probe was controlled by a micromanipulator (MP-225, Butter Instrument). Recordings were usually performed at 400, 600 700 and 800 μm depths in each animal using a pre-amplifier and multiplexer coupled to a USB acquisition card (Intan Technologies).

## Electrophysiology data analysis

Single unit spikes were detected automatically using the klusa Suite (https://github.com/kwikteam/phy). Spikes were then sorted using KlustaKwik spike sorting algorithm (*Harris et al., 2000*) (Klusta, https://github.com/kwikteam/klusta). All posterior data analyses were performed using custom Python scripts. Firing rates were calculated in 25 ms time bins, averaging over 20 sound repetitions.

## Statistics

Data are displayed as mean and S.E.M. Sample size was chosen without explicit power analysis to large enough (n > 4) to allow detection of significance with typical non-parametric tests.

To assess the significance of the relative distribution of functional cell types in V1-projecting neurons and in layer 2/3 neurons, we performed a bootstrap analysis. The null-hypothesis is that both populations have the same distribution of clusters. To simulate this hypothesis, we pooled together all neurons and performed 10,000 random partitions of the pool population into the number of V1-projecting and L2/3 neurons. For each partition, we computed the difference between the fractions of each cluster across the two partitions. This led to distributions of expected fractional differences for each cluster under the null hypothesis. The p-value was computed from the percentile of which the actual observed fractional difference was located.

Significant responses for individual neurons were detected using the non-parametric Wilcoxon rank-sum test. Raw calcium fluorescence traces were subtracted for pre-stimulus level, and averaged over a time window near the response peak. The vector of such responses for different trials was compared to the same computations performed on blank trials (unless responses to two different conditions were compared, such as auditory responses in the dark vs. in a dim light). Unless specified otherwise, we used a statistical threshold of 1% for detecting responding neurons; by definition this means that on average 1% of the neurons *not responding* to the condition would be detected as responding (false positive). Therefore, in all the histogram displays of the fractions of responding neurons (e.g. *Figure 5C,F*) we masked the first 1% as being potentially only false positive detections.

To assess the significance of supra- or sub-linear responses to audio-visual combinations in individual clusters resulting from the clustering of bimodal responses, we used a bootstrap consisting in shuffling the different trial repetitions. The null hypothesis is that the cluster's average bimodal responses can be predicted as the sum of average visual and auditory responses, after baseline has been subtracted. To test this hypothesis, we first computed, for each neuron, individual 'repetitions' of the linear predictions by adding responses to one visual presentation and to one auditory presentation. Then we performed one million shuffles of the labels of 'linear prediction' and 'bimodal response' trials, yielding a distribution of the expected difference between their averages under the null hypothesis. The p-values were computed from the percentile in which the actual nonlinear difference was located.

## Model

Simulations were performed using a rate model with 2 or 3 populations and no synaptic delay. The spiking activity $r_i$ of population $i$ followed the equation:

$$r_i(t) = f\left( b^i_{\text{context}} + g^i_{\text{auditory}}\, s_{\text{auditory}}(t) + g^i_{\text{visual}}\, s_{\text{visual}}(t) + \sum_j g^i_j\, r_j(t-1) \right),$$

Where $t$ is time, $s_{\text{auditory}}$ and $s_{\text{visual}}$ are the auditory and visual input (displaying an exponentially decaying signal and a plateauing signal, respectively), $g^i_{\text{auditory}}$, $g^i_{\text{visual}}$, and $g^i_j$ the connectivities to input and to the other populations (see model schematics in *Figure 7A* for which connections are non-zero). $b^i_{\text{context}}$ is a baseline accounting for other inputs, and whose value can depend on the context ('dark' or 'light'). $f$ is a nonlinear function consisting of two linear segments (*Figure 7A*) that accounts for spiking threshold or further nonlinear (e.g. dendritic) computations. The first segment is not a constant zero accounts for the fact that the other inputs summarized in $b^i_{\text{context}}$ are in fact stochastic and can lead the cell to fire even when its average potential is below threshold.

## Data and software availability

The data that support the findings of this study are freely available at https://www.bathellier-lab.org/downloads or at Dryad, doi:10.5061/dryad.82r5q83. Custom analysis scripts are available in *Source Code 1*.

## Acknowledgements

We thank A Fleischmann, Y Frégnac and J Letzkus for helpful discussions and comments on the manuscript. We thank A Fleischmann for providing CAV2-Cre viruses. This work was supported by the Agence Nationale pour la Recherche (ANR 'SENSEMAKER'), the Marie Curie FP7 Program (CIG 334581), the Human Brain Project (SP3 - WP5.2), the European Research Council (ERC CoG Deepen), the Fyssen foundation, the DIM 'Region Ile de France', the International Human Frontier Science Program Organization (CDA-0064–2015), the Fondation pour l'Audition (Laboratory grant), École Doctorale Frontières du Vivant (FdV) – Programme Bettencourt and the Paris-Saclay University (Lidex NeuroSaclay, IRS iCode and IRS Brainscopes).

## Additional information

### Funding

| Funder | Grant reference number | Author |
|---|---|---|
| Agence Nationale de la Recherche | Retour Postdoc | Deneux Thomas<br>Brice Bathellier |
| Human Frontier Science Program | CDA | Evan R Harrell<br>Brice Bathellier |
| H2020 European Research Council | ERC CoG | Alexandre Kempf<br>Sebastian Ceballo<br>Anton Filipchuk<br>Brice Bathellier |
| Seventh Framework Programme | Marie Curie CiG | Brice Bathellier |
| Fondation pour l'Audition | Lab grant | Evan R Harrell<br>Brice Bathellier |
| École Doctorale Frontières du Vivant (FdV) - Programme Bettencourt | Phd fellowship | Alexandre Kempf |
| Paris-Saclay University | NeuroSaclay Brainscopes | Brice Bathellier |
| Ecole des Neurosciences de Paris | Phd fellowship | Sebastian Ceballo |

The funders had no role in study design, data collection and interpretation, or the decision to submit the work for publication.

## Author contributions

Thomas Deneux, Conceptualization, Data curation, Software, Formal analysis, Investigation, Visualization, Methodology, Writing—original draft, Wrote the paper, Designed all experiments, Carried out all visual cortex aspects of experiments, Collected and analyzed the data, Assisted with preparing the manuscript; Evan R Harrell, Conceptualization, Validation, Investigation, Methodology, Wrote the paper, Designed, carried out chemogenetic and muscimol experiments, Collected and analyzed the data; Alexandre Kempf, Conceptualization, Software, Formal analysis, Investigation, Visualization, Designed, carried out some auditory cortex two-photon imaging experiments, Collected and analyzed some of the data; Sebastian Ceballo, Formal analysis, Investigation, Designed, carried out auditory cortex electrophysiology experiments, Collected and analyzed the data; Anton Filipchuk, Data curation, Investigation, Carried out some auditory cortex two-photon imaging experiments, Collected and pre-processed the data; Brice Bathellier, Conceptualization, Data curation, Formal analysis, Supervision, Funding acquisition, Validation, Investigation, Methodology, Writing—original draft, Project administration, Writing—review and editing, Wrote the paper, Designed all experiments, Performed some two-photon imaging experiments, and analyzed auditory cortex data, Supervised the project

## Author ORCIDs

Thomas Deneux (iD) https://orcid.org/0000-0002-9330-7655
Evan R Harrell (iD) https://orcid.org/0000-0002-4376-9405
Brice Bathellier (iD) https://orcid.org/0000-0001-9211-1960

## Ethics

Animal experimentation: All animal procedures were approved by the French Ethical Committee (authorization 00275.01).

## Decision letter and Author response

Decision letter https://doi.org/10.7554/eLife.44006.021
Author response https://doi.org/10.7554/eLife.44006.022

# Additional files

## Supplementary files

• Source code 1. Custom analysis scripts.
DOI: https://doi.org/10.7554/eLife.44006.016

• Transparent reporting form
DOI: https://doi.org/10.7554/eLife.44006.017

## Data availability

The data that support the findings of this study are freely available at https://www.bathellier-lab.org/downloads or at Dryad, doi:10.5061/dryad.82r5q83. Custom analysis scripts are available as a source code file.

The following dataset was generated:

| Author(s) | Year | Dataset title | Dataset URL | Database and Identifier |
|-----------|------|---------------|-------------|-------------------------|
| Deneux T, Harrell E, Kempf A, Ceballo S, Filipchuk A, Bathellier B | 2019 | Data from: Context-dependent signaling of coincident auditory and visual events in primary visual cortex | https://dx.doi.org/10.5061/dryad.82r5q83 | Dryad Digital Repository, 10.5061/dryad.82r5q83 |

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
