## [Decision Letter]

Thank you for submitting your article "Context-dependent signaling of coincident auditory and visual events in primary visual cortex" for consideration by *eLife*. Your article has been reviewed by three reviewers and the evaluation has been overseen by a Reviewing Editor andAndrew King as the Senior Editor. The following individuals involved in review of your submission have agreed to reveal their identity: Adi Mizrahi (Reviewer #1); Anthony Holtmaat (Reviewer #3).

The reviewers have discussed the reviews with one another and the Reviewing Editor has drafted this decision to help you prepare a revised submission.

Psychophysical data suggest that the auditory and visual systems are intimately connected. This paper is about the circuit mechanisms that support these multimodal interactions. Deneux and colleagues perform an elegant set of experiments characterizing neuronal responses in specific cell types in both auditory and visual cortex to reveal a circuit that could support context-dependent integration of both modalities in the visual cortex. They find that neurons in auditory cortex that project to visual cortex are more likely to be responsive to loud sounds (and to loud sounds getting quieter). They show that these same sounds preferentially modulate the activity of neurons in the visual cortex, but that they do so in a context-dependent and layer-specific manner which depends on the ambient luminance. Some of the data are of high quality and the authors perform a range of controls that lend rigor and confidence to their study. This will be of great interest to a wide audience interested in sensory integration and cortical circuits.

However, the reviewers noted some major concerns that need to be addressed before publication. The reviewers believe that these issues can be addressed in a couple of months.

Conceptual issues:

Specificity for "loud onsets sounds" has not been demonstrated. Although the effects they describe may be stronger for loud onsets than for quiet onsets, the results seem more of a bias within a continuous distribution rather than true specificity.

Another major concern relates to the strong conclusion that the gating of context must occur in the visual cortex instead of in the inputs from the auditory cortex to the visual cortex. This does not seem supported by the data.

These concerns can be addressed by toning down the conclusions in the paper.

Essential revisions:

The concerns listed below require some changes in analysis or data presentation and possibly some experiments that can be done rapidly.

1) The authors state that the majority of V1-projecting neurons in AC originate in L5. However, when they image neurons in AC to monitor sound-evoked responses they compare these responses to L2/3 neurons. What was the rationale behind this comparison? Wouldn't it have been more interesting to compare with the 'overall' population in L5? It is critical to provide images and fluorescence traces in L5 neurons. Along similar lines, the authors do not describe all differences in the response types between V1-projecting AC neurons and the control population. It seems that cluster 7 (OFF-response type) is not present in V1-procting neurons. More histological data needs to be provided to sow which other targets the AC neurons have. This needs to be cleared up.

Also, in the realm of image analysis, the hierarchical clustering should be unpacked in a supplemental figure. How similar are neurons in each cluster? How much variance do the clusters explain etc.

It would have been more convincing to show data from mice where both V1-projecting and those that do not are imaged in the same mice (you already have the tools to do this). Imaging separate neurons in separate experiments (with and without specificity) is less compelling. It’s not absolutely necessary, but if possible, such an experiment (even for a limited number of example) could strengthen these results.

2) The new aspects of the modulation described here relate to the inhibition in darkness vs excitation in light, and the supralinear responses between perceptually matching stimuli. Therefore, the inhibitory responses need to be understood well, especially given that the 'switch' in the model strongly hinges on this finding.

However, the inhibitory effects of sound in darkness are a bit enigmatic and not very well supported by the figures. The authors provide mean deconvolved traces, but these are difficult to digest in the context of these types of responses, as this assumes that there is high baseline activity in V1 in darkness. Either there is a general but very consistent small decrease in all neurons, or a decreased response in a few neurons that are highly active under darkness – but then why the low variance.

For negative responses it is important to exclude the possibilities that technical issues have seeped in. For example, can the authors rule out changes due to vertical movements; and how much does the neuropil signal subtraction affect these responses? 0.7Fnp is a general and accepted rule for neuropil subtraction. However, neutral responses (i.e. no change) might be very sensitive to the 0.7x threshold (especially in densely labeled populations). Could the authors 'play' with these parameters to see how they affect the outcome (e.g. same analysis with and without NP correction or varying thresholds), and convincingly show at the level of individual neurons that their responses were indeed reduced?

Altogether, it is essential to provide example images of groups of neurons (preferably time lapse images) and traces of individual neurons. They should also compare baseline neuronal activity of those neurons that are inhibited versus those who are not, and possibly perform movement (z-plane) correction in images in which they have neurons labeled in red.

3) The authors try and exclude the possibility that a difference in arousal state between the dark and light trials could explain the V1 sign switching. However, it is unclear that closing one eye could simply test the influence of arousal, as this would by itself represent yet a different state of arousal. It is not simply a matter of being in light or darkness, as there are many factors in an experimental setup that determine arousal.

4) In Figure 4, the authors inhibited AC to test whether auditory responses in V1 are caused by AC projections. Where are the statistical comparisons for the DREADD experiments (n=3)? The authors report that this showed a "similar effect, but less robust" as compared to the muscimol experiment. Whereas this trend might be true for the experiments under light, this remains inconclusive for the DREADD experiments in darkness since in one animal the average responses were drastically reduced. The authors should report the statistical comparison for the DREADD experiment and increase the n if they feel that this addition is necessary to support the conclusions.

5) Why opt out of showing 'sound only' averages in Figure 6D? For unimodal stimuli was there any background stimulus in either modality (e.g. for sound only, dim light; and for light only, white noise)? If not, why is the sound-only inhibitory response in V1 (as seen in Figure 5) not reproduced in Figure 6?

6) The results of what they call context are the most interesting part of the paper. The claims of effect-specificity to loud onset sounds and looming stimuli are too strong. After all, they only tested a limited amount of stimuli, both auditory and visual. And even within this limited set, the effects are not binary.

7) The value of the minimal model of Figure 7 is unclear. Reproducing the empirical results with a model is a good starting point but eventually it has to provide something more (e.g. some hypotheses to test in the context of mechanism). Do they suggest specific biophysical mechanisms of the neurons are involved? This should be clarified.

8) The authors' major conclusion is that (from the Abstract) "a small number of layer 1 interneurons gates this cross-modal information flow". They then design a model to demonstrate how this might work through the application of distinct gain conditions and a non-linear threshold. However, another possibility (as the authors acknowledge in the Discussion section) is that the gating might occur in the auditory cortex such that the inputs to the L1 population are only active in the dark. Evidence that the A1->V1 population is insensitive to ambient light conditions would add significant support to the authors assumed model.

9) The authors' model suggests that ambient light alters the excitability (and therefore gain) of L1 interneurons. This is a hypothesis that the authors could test by measuring F in baseline conditions (in the absence of auditory stimulation) in the light and dark. Evidence that the L1 neurons that are driven only in the dark have higher baseline F in the dark (while other less selective L1 interneurons do not show such strong modulation) would significantly strengthen the authors' argument.

10 The authors' proposed model assumes a functionally homogeneous input to V1. However, while the majority of V1->A1 neurons prefer down-ramp sounds, there are still a significant number that respond to up-ramps. In fact, up-ramp sounds are sufficient to drive suppression in the dark, though unlike down-ramp sounds do not evoke either excitation or inhibition in the light (at least not on average as shown in Figure 3B). This suggests that down-ramp preferring neurons may only contact L1 interneurons and not provide direct excitation to L2/3. Thus, the authors should make it clear that there are anatomical specializations that might also support the observed gating.

---

## [Author Response]

Psychophysical data suggest that the auditory and visual systems are intimately connected. This paper is about the circuit mechanisms that support these multimodal interactions. Deneux and colleagues perform an elegant set of experiments characterizing neuronal responses in specific cell types in both auditory and visual cortex to reveal a circuit that could support context-dependent integration of both modalities in the visual cortex. They find that neurons in auditory cortex that project to visual cortex are more likely to be responsive to loud sounds (and to loud sounds getting quieter). They show that these same sounds preferentially modulate the activity of neurons in the visual cortex, but that they do so in a context-dependent and layer-specific manner which depends on the ambient luminance. Some of the data are of high quality and the authors perform a range of controls that lend rigor and confidence to their study. This will be of great interest to a wide audience interested in sensory integration and cortical circuits.However, the reviewers noted some major concerns that need to be addressed before publication. The reviewers believe that these issues can be addressed in a couple of months.

To address these concerns, we have provided three new sets of experiments (imaging in layer 5, electrophysiology in layer 5, and imaging of V1 projecting cells in AC in lit vs dark conditions). We also performed the requested complementary analysis

Conceptual issues:Specificity for "loud onsets sounds" has not been demonstrated. Although the effects they describe may be stronger for loud onsets than for quiet onsets, the results seem more of a bias within a continuous distribution rather than true specificity.

We agree that the results describe a different bias in the distribution of sound response types, rather than a strict specificity.

We thus carefully checked the text and made several changes in the Abstract and main text to avoid conveying the impression that there is strict specificity. For example, we have replaced ‘specific’ by ‘preferential’, wherever meaningful, or described, in accordance with the data, that loud onsets are a dominant feature among other features encoded in V1-projecting cells. Below are a few examples of modified text:

Abstract: “Using two-photon calcium imaging in identified cell types in awake, head-fixed mice, we show that, among the basic features of a sound envelope, loud sound onsets are a dominant feature coded by the auditory cortex neurons projecting to primary visual cortex (V1). […] when sound input coincides with a visual stimulus, visual responses are boosted in V1, most strongly after loud sound onsets.”

Introduction: “and [we] showed that, V1-projecting neurons are predominantly tuned to loud onsets while other tested envelope features are less prominently represented, at least when comparing with supragranular cortical layers that lack V1-projecting cells.…this mechanism also allows for a boosting of visual responses together with auditory events, particularly when those include loud sound onsets, increasing the saliency of visual events that co-occur with abrupt sounds.”

Results, subsection “Abrupt sound onsets are a dominant feature encoded by V1-projecting neurons”

Discussion section: “First, we showed that V1-projecting neurons have a different bias in the distribution of sound envelope features they encode, as compared to layer 2/3 neurons in AC and also, although to a lesser extent, as compared to layer 5 in which they are located. In particular, the V1’projecting neuron population predominantly encodes loud sound onsets.”

Plus a larger more detailed paragraph describing the changes in response type distribution for all cell subpopulations.

Another major concern relates to the strong conclusion that the gating of context must occur in the visual cortex instead of in the inputs from the auditory cortex to the visual cortex. This does not seem supported by the data.

We have done new experiments, recording V1 projecting cells in auditory cortex in light and dark conditions. These experiments show that the illumination context does not modulate activity of auditory cortex neurons. Thus, the contextual gating or modulation must happen in V1.

The results are shown in Figure 3F. Note that we have used this new dataset for the clustering analysis of V1 projecting AC neurons in Figure 1 which is thus also changed (and extended with data from AC layer 5).

These concerns can be addressed by toning down the conclusions in the paper.

We have toned down the conclusions related to specificity of loud onset. See our response to the first point above.

Essential revisions:The concerns listed below require some changes in analysis or data presentation and possibly some experiments that can be done rapidly.1) The authors state that the majority of V1-projecting neurons in AC originate in L5. However, when they image neurons in AC to monitor sound-evoked responses they compare these responses to L2/3 neurons. What was the rationale behind this comparison? Wouldn't it have been more interesting to compare with the 'overall' population in L5? It is critical to provide images and fluorescence traces in L5 neurons. Along similar lines, the authors do not describe all differences in the response types between V1-projecting AC neurons and the control population. It seems that cluster 7 (OFF-response type) is not present in V1-procting neurons. More histological data needs to be provided to sow which other targets the AC neurons have. This needs to be cleared up.

We failed to achieve simultaneous imaging of V1 projecting and non-projecting cells in AC (see below), a separation which is actually hard to guaranty with viral techniques (partial infection). Thus, we had focused on a comparison between L2/3 (that contains almost no V1 projecting cells) in AC and identified V1 projecting cells. But indeed this leaves open the question whether L2/3 and L5 cells encode a different distribution of features. We have thus now provided in Figure 1 a quantification of the distribution of sound envelop features in upper L5. It shows, among other differences, an increase of the fraction of loud ON responses compared to layer 2/3. We clarified the conclusions by emphasizing the dominance of loud ON response in V1 projecting neurons (half of the cells), a situation different from in layer 2/3, but maybe partially inherited from an enrichment of loud ON responses in layer 5.

We observe differences in the distribution of responses type across all three datasets. For example, as the referees point out, quiet OFF neurons (former cluster 7) are generally very rare (see also Deneux et al., 2016) and are thus not seen in in all dataset, possibly because it was not picked up by the clustering given their sparseness. We provide now a small paragraph on these discrepancies in the Discussion:

“The shifts in the distribution of sound envelope features that we observed across layers and cell subpopulations is an interesting case of coding bias in different neural populations of auditory cortex (Figure 1). As observed earlier, in layer 2/3, there is a balance between the main neuron types that signal the beginning and the end of sounds with equivalent fractions of neurons responding to quiet or loud onsets and loud offsets (Figure 1F, note that ‘tonic+loud OFF’ neurons are included in this count. Quiet offset neurons are rare and not robustly detected across datasets. Detection of tonic responses by clustering is also variable.). In upper layer 5, the distribution becomes more biased towards loud onsets at the expense of offset neurons, a trend that is further increased in the layer 5-located V1-projecting cells in which half of the neurons respond to loud onsets and 10% of the cells to loud offsets.”

Histological data showing which other targets the AC neurons have is freely available online on the website of the Allen Brain Institute and of the mouse i-connectome project (www.mouseconnectome.org). We did not feel we can provide data of better quality during the time frame of the review process.

Also, in the realm of image analysis, the hierarchical clustering should be unpacked in a supplemental figure. How similar are neurons in each cluster? How much variance do the clusters explain etc.

In this revision we have unpacked the clustering of the 3 extended/new dataset by:

a) Allowing more overclustering in order to show more of the variability that appears inside the functional clusters we identified. We have also used more cluster labels, similar to our previous study on ramping sounds (Deneux et al., 2016).

b) we have provided response correlation matrices from which variability inside clusters can be visually estimated (see figure 1 and Figure 1—figure supplement 1)

c) we have measured the fraction of explained variance (about 50% in all cases) and provided the values in the result section.

It would have been more convincing to show data from mice where both V1-projecting and those that do not are imaged in the same mice (you already have the tools to do this). Imaging separate neurons in separate experiments (with and without specificity) is less compelling. It’s not absolutely necessary, but if possible, such an experiment (even for a limited number of example) could strengthen these results.

Originally, we had started our experiments with this approach, labeling V1 projecting neurons with TdTomato, while AAV1-syn-GCAMP6s was used to label broadly the neurons of AC. However, it turned out that GCAMP6s labelling was clearly weaker in the TdT labelled neurons than in the rest of the layer 5 population. Hence, it was not really possible to image them, because their fluorescence was covered by the fluorescence of their neighbors (neuropil signal). This effect was magnified by the slightly decreased spatial resolution and contrast of imaging data in Layer 5 obtained with the bulk AAV1-syn-GCAMP6 injection strategy that also tends to label L2/3 (thus increasing the contribution of out-of-focus light at large depths). Together these technical limitations prevented us from providing data in which cells from the same field of view are imaged and sorted according to their projection target. Note also that absence of retrograde labelling does not indicate absence of projection to V1.

2) The new aspects of the modulation described here relate to the inhibition in darkness vs excitation in light, and the supralinear responses between perceptually matching stimuli. Therefore, the inhibitory responses need to be understood well, especially given that the 'switch' in the model strongly hinges on this finding.However, the inhibitory effects of sound in darkness are a bit enigmatic and not very well supported by the figures. The authors provide mean deconvolved traces, but these are difficult to digest in the context of these types of responses, as this assumes that there is high baseline activity in V1 in darkness. Either there is a general but very consistent small decrease in all neurons, or a decreased response in a few neurons that are highly active under darkness – but then why the low variance.

As we now show in Figure 3—figure supplement 1 panel B, the inhibition is consistent and wide spread across neurons. This explains the low variance.

For negative responses it is important to exclude the possibilities that technical issues have seeped in. For example, can the authors rule out changes due to vertical movements; and how much does the neuropil signal subtraction affect these responses? 0.7Fnp is a general and accepted rule for neuropil subtraction. However, neutral responses (i.e. no change) might be very sensitive to the 0.7x threshold (especially in densely labeled populations). Could the authors 'play' with these parameters to see how they affect the outcome (e.g. same analysis with and without NP correction or varying thresholds), and convincingly show at the level of individual neurons that their responses were indeed reduced?

As we now show in Figure 3—figure supplement 1 panel A, the inhibition is almost unchanged in the absence of neuropil correction as compared to -0.7Fnp. Thus, the neuropil correction does not introduce supplementary inhibition. Excessive neuropil signals correction (-1*Fnp) lowers the basal fluorescence and thereby increase ΔF/F values, but inhibition is preserved.

Altogether, it is essential to provide example images of groups of neurons (preferably time lapse images) and traces of individual neurons. They should also compare baseline neuronal activity of those neurons that are inhibited versus those who are not, and possibly perform movement (z-plane) correction in images in which they have neurons labeled in red.

We have done efficient z-plane correction using the green channel. We do not have simultaneous imaging of the red channel in any dataset (the red image was acquired before at a different wavelength), so we cannot use the red-labelled neurons to perform horizontal motion artefact correction.

We now provide in Figure 3—figure supplement 1 panel C, single trial raw traces and spontaneous activity for representative sample neurons (inhibited, excited by sounds). The inhibitory example shows that inhibition is visible in single trials although not systematically. Together with this inhibition we observe a decrease in the occurrence of occasional transients. This suggests that there is a basal level of activity and that inhibition is revealed as a modulation of this activity level. To validate this idea, we run simulations of GCAMP6s signals based on published parameters for transients resulting from a single spike and supposing basal Poisson firing (different rates were tested). We simulated an inhibition by removing spikes of the Poisson train in a 1s time bin. We show (Figure 3—figure supplement 1 panel D) that resulting calcium signals and their trial average are highly compatible with signals observed in vivo for firing rate as low as 1Hz. Hence, observed inhibition likely results from a consistent modulation of a low basal firing rate, compatible with published basal firing rate values in awake cortex (e.g. loose patch recordings indicate a distribution around 2-3Hz Hromadka, Zador et al., 2008).

This conclusion is stated in the Results section “The observed inhibition was seen consistently across cells and sound presentations, likely corresponding to a transient decrease in basal firing rates (Figure 3—figure supplement 1).”

3) The authors try and exclude the possibility that a difference in arousal state between the dark and light trials could explain the V1 sign switching. However, it is unclear that closing one eye could simply test the influence of arousal, as this would by itself represent yet a different state of arousal. It is not simply a matter of being in light or darkness, as there are many factors in an experimental setup that determine arousal.

We agree that this experiment does not fully address the question of arousal. We now moved the results in Figure 3—figure supplement 2, and together with a sentence that does not mention arousal: “Interestingly, hiding the contralateral eye was sufficient to obtain the same inhibition as in darkness (Figure 3—figure supplement 2), suggesting a possible role of direct unilateral visual inputs in the context dependence of auditory responses in V1”

4) In Figure 4, the authors inhibited AC to test whether auditory responses in V1 are caused by AC projections. Where are the statistical comparisons for the DREADD experiments (n=3)? The authors report that this showed a "similar effect, but less robust" as compared to the muscimol experiment. Whereas this trend might be true for the experiments under light, this remains inconclusive for the DREADD experiments in darkness since in one animal the average responses were drastically reduced. The authors should report the statistical comparison for the DREADD experiment and increase the n if they feel that this addition is necessary to support the conclusions.

The referee is right to point out that the number of mice used in the DREADD experiment is too low for accurate statistical analysis across animals. Yet the observations of DREADD effects are quite robust for the recorded population if one considers variability across sound repeats, as displayed in the standard error of the average response traces. We thus now provide in Figure 4C the statistics across n=20 independent repeats of the stimulus presentations for the entire population of neurons recorded. This shows high significance. For consistency, the variations across mice are shown in Figure 4—figure supplement 1, together with a statement that DREADD silencing is less robust across animals as expected from variations in targeting efficiency in the double viral injection approach, which likely touches only a subset of all neurons projecting to A1 (virus efficiency and tropism; also the CAV-Cre virus is injected in a single spot V1).

As the DREADD experiments essentially corroborates the muscimol experiments and are thus not crucial for our conclusions, we did not increase the n.

New text: “We also used targeted chemogenetic inhibition [...] This produced a decrease of sound responses which was consistent across sound repetitions for the imaged neural population (Figure 4C). However, probably because this strategy impacts an incomplete subset of all V1-projecting neurons, the effects were smaller and more variable across experiments (Figure 4—figure supplement 1). “

5) Why opt out of showing 'sound only' averages in Figure 6D? For unimodal stimuli was there any background stimulus in either modality (e.g. for sound only, dim light; and for light only, white noise)? If not, why is the sound-only inhibitory response in V1 (as seen in Figure 5) not reproduced in Figure 6?

The sound only responses in muscimol and saline (both in light and dark conditions) were recorded in the same experiment and are already shown in Figure 4. Adding them in Figure 6D would mean overcrowding the figure with 4 more curves in each plot (6 curves in total), this is the reason why we did not reproduce them.

6) The results of what they call context are the most interesting part of the paper. The claims of effect-specificity to loud onset sounds and looming stimuli are too strong. After all, they only tested a limited amount of stimuli, both auditory and visual. And even within this limited set, the effects are not binary.

We have toned down the conclusions throughout the paper to avoid conveying the wrong impression that the effects are just restricted to Loud onset sounds (see our response above).

7) The value of the minimal model of Figure 7 is unclear. Reproducing the empirical results with a model is a good starting point but eventually it has to provide something more (e.g. some hypotheses to test in the context of mechanism). Do they suggest specific biophysical mechanisms of the neurons are involved? This should be clarified.

After describing our model, we now propose a biophysical implementation for the two key aspects:

- For the modulation of a subpopulation of L1 interneurons:

“Biophysically, this mechanism requires a tonic inhibitory drive to the identified subset of L1 interneurons which could be provided by another inhibitory population. Interestingly, we observed that the subpopulation of L1 interneurons that are less responsive to sounds in light also have a smaller baseline fluorescence in light, an effect that is not observed for other L1 interneurons (Figure 7B). This could be a possible manifestation of the tonic drive hypothesized in our model.”

- For the low and high gain behavior of L2/3 neurons:

“While less conventional than a regular spike threshold, this mechanism could be biophysically implemented with non-linear dendritic processing. Apical dendrites of layer 2/3 pyramidal cells are known to produce large calcium spikes which boost somatic output when somatic activity coincides with excitatory inputs arriving in the apical tree (Larkum et al., 2007). As AC axon terminals are concentrated in layer 1 where they likely contact the apical dendrites of pyramidal cells, sound-driven boosting of visual responses could rely on this dendritic phenomenon.”

8) The authors' major conclusion is that (from the Abstract) "a small number of layer 1 interneurons gates this cross-modal information flow". They then design a model to demonstrate how this might work through the application of distinct gain conditions and a non-linear threshold. However, another possibility (as the authors acknowledge in the Discussion section) is that the gating might occur in the auditory cortex such that the inputs to the L1 population are only active in the dark. Evidence that the A1->V1 population is insensitive to ambient light conditions would add significant support to the authors assumed model.

We have done new experiments, recording V1 projecting cells in auditory cortex in light and dark conditions. These experiments show that illumination context does not modulate activity of auditory cortex neurons. Therefore, we continue to believe that this contextual gating or modulation happens in V1.

The results are shown in Figure 3F and commented in the result section: “We then wondered whether the context-dependence was a property arising in the circuit of V1 or if it is due to a modulation of auditory inputs by the light context. We imaged sound responses in V1-projecting neurons of auditory cortex specifically labelled using the CAV-Cre retrograde virus approach (see Figure 1). We observed no response modulation between the dark and lit conditions (Figure 3F), indicating that context-dependent modulation arises in V1.”

9) The authors' model suggests that ambient light alters the excitability (and therefore gain) of L1 interneurons. This is a hypothesis that the authors could test by measuring F in baseline conditions (in the absence of auditory stimulation) in the light and dark. Evidence that the L1 neurons that are driven only in the dark have higher baseline F in the dark (while other less selective L1 interneurons do not show such strong modulation) would significantly strengthen the authors' argument.

We have done this analysis. Indeed, L1 interneurons that are responding less in the light display also a higher baseline in the dark, in line with our model. We now show this result in an inset of Figure 7 and describe it in the Results section:

“provided that they are close to activation threshold in the dark and that inhibition by visual inputs brings them well below threshold in lit condition, interneurons will be less active in the light than in the dark. […] Biophysically, this mechanism requires a tonic inhibitory drive to the identified subset of L1 interneurons which could be provided by another inhibitory population. Interestingly, we observed that the subpopulation of L1 interneurons that are less responsive to sounds in light also have a smaller baseline fluorescence in light, an effect that is not observed for other L1 interneurons (Figure 7B). This could be a possible manifestation of the tonic drive hypothesized in our model”

10) The authors' proposed model assumes a functionally homogeneous input to V1. However, while the majority of V1->A1 neurons prefer down-ramp sounds, there are still a significant number that respond to up-ramps. In fact, up-ramp sounds are sufficient to drive suppression in the dark, though unlike down-ramp sounds do not evoke either excitation or inhibition in the light (at least not on average as shown in Figure 3B). This suggests that down-ramp preferring neurons may only contact L1 interneurons and not provide direct excitation to L2/3. Thus, the authors should make it clear that there are anatomical specializations that might also support the observed gating.

This is a good point, which we added to the Discussion section. “As a result of these fine adjustments of feature distributions, layer 2/3 neurons in V1 are more impacted by abrupt sounds with loud onsets (e.g. down-ramps) than by sounds with softer onsets (e.g. up-ramps). This asymmetry is particularly striking for excitatory effects seen in light or with visual stimuli (Figures 3B and 6), but less pronounced for inhibitory effects in the dark (Figures 3B). It is thus possible that the projection targets also depend on encoded features. For example, L2/3 V1 pyramidal cells may receive projections almost exclusively from AC loud onset neurons while interneurons providing inhibition in the dark still receive a significant fraction of quiet onset and loud offset inputs.”